# COCO-Counterfactuals: Automatically Constructed Counterfactual Examples for Image-Text Pairs

**Tiep Le**[*]
Intel Labs
tiep.le@intel.com

**Vasudev Lal**
Intel Labs
vasudev.lal@intel.com

**Phillip Howard**[*]
Intel Labs
phillip.r.howard@intel.com

## Abstract

Counterfactual examples have proven to be valuable in the field of natural language processing (NLP) for both evaluating and improving the robustness of language models to spurious correlations in datasets. Despite their demonstrated utility for NLP, multimodal counterfactual examples have been relatively unexplored due to the difficulty of creating paired image-text data with minimal counterfactual changes. To address this challenge, we introduce a scalable framework for automatic generation of counterfactual examples using text-to-image diffusion models. We use our framework to create COCO-Counterfactuals, a multimodal counterfactual dataset of paired image and text captions based on the MS-COCO dataset. We validate the quality of COCO-Counterfactuals through human evaluations and show that existing multimodal models are challenged by our counterfactual image-text pairs. Additionally, we demonstrate the usefulness of COCO-Counterfactuals for improving out-of-domain generalization of multimodal vision-language models via training data augmentation. We make our code[2] and the COCO-Counterfactuals dataset[3] publicly available.

## 1 Introduction

While vision and language models have achieved remarkable performance improvements in recent years, out-of-domain (OOD) generalization remains a challenge for even the best models, which typically exhibit much lower performance in zero-shot evaluation settings than on withheld in-domain test sets. This has often been attributed to spurious correlations between non-causal features and labels in datasets which can be exploited during training as shortcuts to achieving artificially high in-domain performance (Geirhos et al., 2020). For example, image recognition models often learn to utilize spurious features in the backgrounds of images when trained for classification on datasets such as ImageNet (Singla and Feizi, 2021; Xiao et al., 2020).

Augmenting training datasets with counterfactuals, which study the impact on a response variable following a change to a causal feature, has been previously proposed as a strategy for countering this effect in NLP models (Levesque et al., 2012; Kaushik et al., 2019). Motivated by concepts in causal learning (Feder et al., 2022), these methods typically form counterfactual examples by making minimal edits to an input text such that a corresponding label or attribute of the text (e.g., sentiment)

---

[*]Equal contribution
[2]https://github.com/IntelLabs/multimodal_cognitive_ai/tree/main/COCO-Counterfactuals
[3]https://huggingface.co/datasets/Intel/COCO-Counterfactuals

37th Conference on Neural Information Processing Systems (NeurIPS 2023) Track on Datasets and Benchmarks.

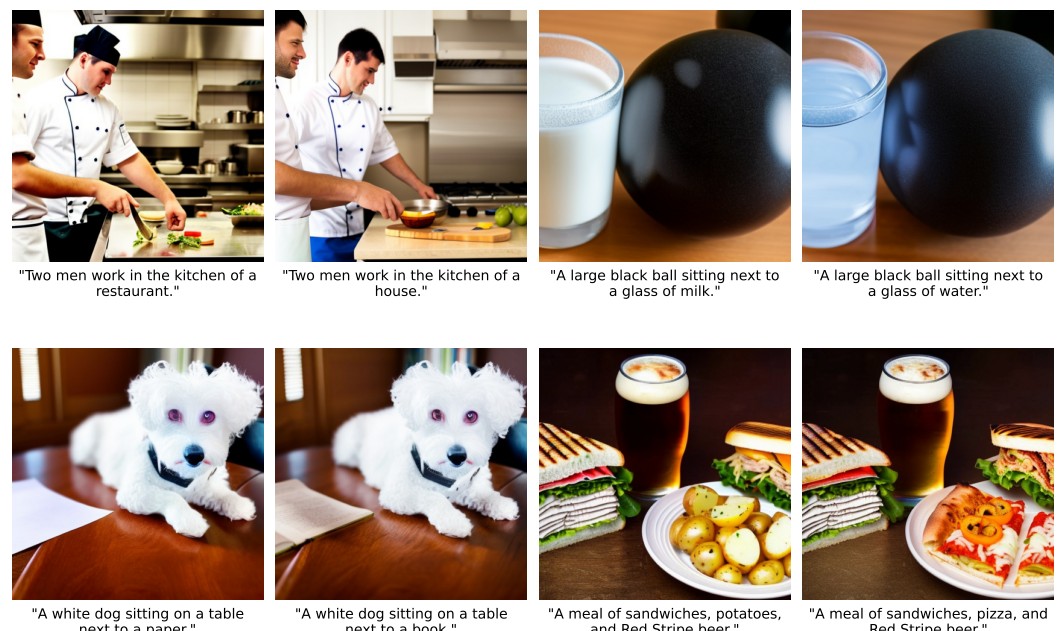

"Two men work in the kitchen of a restaurant."
"Two men work in the kitchen of a house."
"A large black ball sitting next to a glass of milk."
"A large black ball sitting next to a glass of water."

"A white dog sitting on a table next to a paper."
"A white dog sitting on a table next to a book."
"A meal of sandwiches, potatoes, and Red Stripe beer."
"A meal of sandwiches, pizza, and Red Stripe beer."

Figure 1: Examples of COCO-Counterfactuals, our minimal-edit counterfactuals dataset for images with paired text captions.

is changed. Training models with counterfactual examples therefore provides a strong inductive bias against learning spurious correlations in datasets, leading to greater robustness and improved generalization on OOD data (Eisenstein, 2022; Vig et al., 2020) as well as enabling better domain adaptation in low resource settings (Calderon et al., 2022).

Despite its success in the realm of NLP, the application of counterfactual data augmentation to multimodal vision-language models has largely remained unexplored, mainly due to low-resource settings involving multimodal data and challenges associated with creating paired counterfactual examples spanning multiple modalities. For example, consider the task of creating counterfactual examples for a multimodal dataset containing images with associated text captions. Creating a counterfactual to a given image-text example requires not only minimally editing a casual feature in the text caption, but also making a corresponding minimal edit to the image which ideally modifies only the changed causal feature while preserving other spurious features from the original image.

Collecting such counterfactual examples from existing image datasets is infeasible due to the massive variation in natural images that can accurately depict even identical text captions. While manual creation of counterfactual examples by humans is an option that has been employed previously for NLP (Kaushik et al., 2019; Gardner et al., 2020), this approach suffers from a lack of scalability due to the high cost of human labor, which would be compounded even further for multimodal counterfactuals due to the need for both text and image editing skills. Given these challenges, how can paired image-text counterfactual examples be created at the scale needed for effective model evaluation and data augmentation?

We address this problem by introducing a novel data generation pipeline for automatically creating multimodal counterfactual examples using text-to-image diffusion models. Our approach minimally edits captions from an existing image-text dataset and then leverages Stable Diffusion (Rombach et al., 2021) with cross-attention control (Hertz et al., 2022) to generate pairs of images with minimal differences (i.e., isolated to the counterfactual change). We employ our data generation pipeline to create at scale **COCO-Counterfactuals** (Figure 1), a counterfactual variant of the MS-COCO dataset (Lin et al., 2014).

We validate the quality of COCO-Counterfactuals using human evaluations and conduct zero-shot experiments showing that state-of-the-art multimodal models are challenged by our generated counterfactual examples. Our additional experiments show that training CLIP (Radford et al., 2021)

on COCO-Counterfactuals improves its performance on multiple OOD datasets, including zero-shot tasks not seen during training. We make COCO-Counterfactuals and the code for our counterfactual data generation pipeline publicly available under the CC BY 4.0 License.

## 2 Related Work

### 2.1 Counterfactual Examples for NLP

Counterfactual data augmentation has been shown to improve the robustness of models across a wide range of problem domains in NLP. Kaushik et al. (2019) demonstrated that human-authored counterfactuals pose a significant challenge for existing models and that augmenting training datasets with counterfactual examples improves sentiment analysis and Natural Language Inference classifiers. Gardner et al. (2020) similarly used human experts to author minimally-edited contrast example sets for 10 NLP datasets and showed that model performance evaluated on them drops substantially.

A number of approaches have been proposed to move beyond reliance on human authors towards automated methods for generating counterfactual examples. Wang and Culotta (2021) and Yang et al. (2021) automatically construct counterfactual examples by identifying and removing or replacing potentially causal words. Howard et al. (2022) introduce a framework for generating looser counterfactuals which allow larger edits of original examples, resulting in more natural and linguistic counterfactual examples. Other semi-automated methods have been proposed to generate counterfactual examples while still relying on human input or labeling (Wu et al., 2021). To the best of our knowledge, none of these existing approaches for automatic counterfactual generation have been extended to multimodal image-text datasets.

### 2.2 Image Benchmarks for Measuring Spurious Correlations

Several image datasets have been proposed as benchmarks for measuring the degree to which models have learned to rely on spurious correlations during training. *CelebA Hair Color* (Liu et al., 2015) is a binary image classification dataset that labels whether a person depicted has a blonde hair color, which is spuriously correlated with gender. Sagawa et al. (2019) constructed the *Waterbirds* dataset by cropping images of landbirds or seabirds onto land and sea backgrounds, resulting in a binary classification task for bird type (i.e., landbird or seabird) in the presence of spurious correlations with the background. *Colored MNIST* (Arjovsky et al., 2019) artificially imposes colors on the MNIST handwritten digits dataset, where the color is spuriously correlated with the class label. Lynch et al. (2023) uses text-to-image models to generate *Spawrious*, an image classification dataset of four dog breeds spuriously correlated with six background locations. Unlike COCO-Counterfactuals, these datasets are limited only to image classification over a small number of labels and are primarily suited for evaluating model robustness as opposed to training data augmentation.

Thrush et al. (2022) introduced *Winoground*, an image-text dataset aimed at measuring visio-linguistic compositionality. Given two images and two captions which have the same words but in different order, the task is to correctly match each caption to its corresponding image. While their paired image-text examples can be viewed as counterfactuals, they focus only on edits to word order and rely on humans to create a dataset aimed specifically at evaluating compositionality. In contrast, our method automatically generates counterfactual examples with word content changes while also preserving non-causal spurious features across paired counterfactual images.

*FOIL-COCO* (Shekhar et al., 2017) contains 'foil' captions with a single change to the original MS-COCO caption to invalidate it for the accompanying image. They show that vision and language models struggle to correctly classify captions, detect the edited word, and correct the foiled caption. Our image-text counterfactuals similarly create 'foil' captions to MS-COCO captions, but goes further by also creating paired images which differ only according to how the caption was edited.

### 2.3 Data Augmentation with Synthetic Images Generated from Text-to-image Models

Motivated by recent advances in text-to-image diffusion models (Nichol et al., 2021; Rombach et al., 2021; Saharia et al., 2022; Ramesh et al., 2022), data augmentation with synthetically-generated images has emerged as a growing topic of interest. He et al. (2022) showed that images generated by GLIDE (Nichol et al., 2021) for specific classes in image recognition datasets can be used for

training to improve performance on the corresponding image classification tasks. Trabucco et al. (2023) perform image-to-image transformations for data augmentation using text-to-image diffusion models, observing improvements in few-shot image classification performance. Vendrow et al. (2023) represent class labels from image recognition datasets as custom tokens in the vocabulary of a text-to-image diffusion model, enabling them to generate images of objects from the original dataset under different domain shifts. While our data generation pipeline also leverages text-to-image diffusion models, our approach differs from prior work in our focus on producing minimal changes to paired image-text data in both the vision and language modalities.

## 3 COCO-Counterfactuals

We detail our data generation methodology for creating COCO-Counterfactuals (COCO-CFs), a synthetic multimodal counterfactual dataset of paired image and text captions based on the MS-COCO dataset (Lin et al., 2014). While we showcase our methodology by generating and releasing the COCO-Counterfactuals dataset, our approach can be applied to automatically construct multimodal counterfactuals for any dataset containing image captions.[4]

### 3.1 Creating Counterfactual Captions

Given an original image caption $C_o$, our first task is to create a corresponding counterfactual caption $C_c$ which alters a subject of $C_o$ while preserving most of its original details. The altered subject represents the changed causal feature in our counterfactual example while the remaining preserved details from the original caption can be viewed as potentially spurious correlated features.

To alter a subject of $C_o$, we first identify all nouns using NLTK (Bird et al., 2009) as candidate words for substitution[5] For each of the $i \in \{1, .., n\}$ identified nouns, we create 10 candidate counterfactual captions by replacing only the $i$-th noun in $C_o$ with the [MASK] token and retrieving the top-10 most probable replacements via masked language modeling (MLM)[6]. This produces a total of $n \times 10$ candidate counterfactual captions, which we then filter to retain only those in which the substituted word is also a noun.

Our aim is to substitute nouns with alternative words that represent different subjects, and yet still maintain ontological similarity to the original noun. Hence, we use a pre-trained sentence similarity model[7] to measure the similarity between each candidate counterfactual caption and the original caption $C_o$, keeping only those candidates which have a sentence similarity within the range $(0.8, 0.91)$. Finally, we use GPT-2 to score the perplexity of all candidates which remain after filtering and choose the candidate having the lowest perplexity as our counterfactual caption $C_c$.

### 3.2 Generating Counterfactual Images

After creating a counterfactual caption $C_c$, our next task is to generate synthetic images $I_o^s$ and $I_c^s$ from the original caption $C_o$ and counterfactual caption $C_c$ (respectively). Ideally we would like $I_o^s$ and $I_c^s$ to differ only in terms of the noun which was replaced in $C_o$ to produce $C_c$, thereby enabling the changed causal feature to be learned in the presence of other potentially spurious correlated features (i.e., the unchanged details between $C_o$ and $C_c$). However, this is a challenge for existing text-to-image generation models as minor changes to a text prompt can produce significantly different images. For instance, prompting Stable Diffusion with the captions "A small child lounges with a *remote* in his hand" and "A small child lounges with a *toy* in his hand" may produce images that differ not only in the object that the child is holding, but also in other details such as his facial features, the manner in which he is laying, the color of his clothes, and the image background.

To address this issue, Hertz et al. (2022) proposed a methodology called Prompt-to-Prompt which injects cross-attention maps during the diffusion process to control the attention between certain pixels and tokens of the prompt during denoising steps. This enables separate generations from

---

[4] Appendix B.1 details hyper-parameters and pre-trained models used to generate COCO-Counterfactuals.

[5] In this work, we focus on counterfactual captions that are derived from altering a noun from original captions. We leave the investigation of altering words of other types such as verbs and adjectives for future work.

[6] We use RoBERTa-base for MLM

[7] https://huggingface.co/sentence-transformers/all-MiniLM-L6-v2

With Prompt-to-Prompt             Without Prompt-to-Prompt

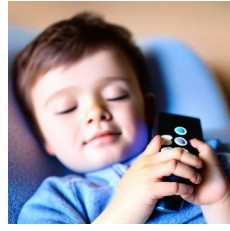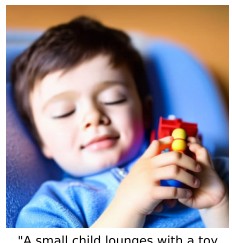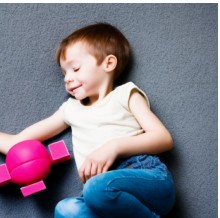

"A small child lounges with a remote in his hand."  "A small child lounges with a toy in his hand."    "A small child lounges with a remote in his hand."  "A small child lounges with a toy in his hand."

Figure 2: Examples of COCO-Counterfactuals generated with Prompt-to-Prompt (left) and without (right). Prompt-to-prompt enables us to extend the principle of minimal-edit text counterfactuals to the visual domain, isolating image differences to only the changed causal feature.

text-to-image diffusion models to maintain many of the same image details while isolating their differences according to how the text prompts differ. An example of counterfactual image-text pairs $(C_o, I_o^s)$ and $(C_c, I_c^s)$ generated with and without prompt-to-prompt is shown in Figure 2, illustrating how Prompt-to-Prompt enables the principle of minimal text edits for NLP counterfactuals to be extended to image generation.

Brooks et al. (2023) noted that making different changes to images may require varying the parameter $p$ in Prompt-to-Prompt, which controls the number of denoising steps with shared attention weights. For example, changes that require more substantial structural modifications to the image may necessitate less overall similarity between the resulting images and thus fewer shared attention weights. We therefore adopt their proposed approach of over-generating 100 image pairs with Prompt-to-Prompt by randomly sampling values of the parameter $p \sim U(0.1, 0.9)$[8]. The resulting 100 image pairs are filtered using CLIP (Radford et al., 2021) to ensure a minimum cosine similarity of 0.2 between the encoding of each caption and its corresponding generated image, with the best image pair $(I_o^s, I_c^s)$ chosen from those which remain according to the directional similarity in CLIP space (Gal et al., 2022):

$$\text{CLIP}_{dir} = \frac{(E_T(C_c) - E_T(C_o)) \cdot (E_I(I_c^s) - E_I(I_o^s))}{||E_T(C_c) - E_T(C_o)|| \; ||E_I(I_c^s) - E_I(I_o^s)||} \tag{1}$$

where $E_T$ and $E_I$ are CLIP's text and image encoders (respectively). The $\text{CLIP}_{dir}$ metric measures the consistency in changes between the two images $(I_o^s, I_c^s)$ and their corresponding captions $(C_o, C_c)$. Thus, selecting images with a higher $\text{CLIP}_{dir}$ improves the overall quality of our generated counterfactuals via greater consistency between the alterations made in both modalities.

### 3.3 Generating COCO-Counterfactuals from MS-COCO

We apply our counterfactual caption and image generation pipeline described above to create the COCO-Counterfactuals dataset. Specifically, we generate candidate counterfactual captions for 25,014 original MS-COCO captions[9][10], keeping only the best candidate counterfactual for each original caption that meets our filtering criteria. This produced a total of 24,508 original & counterfactual caption pairs $(C_o, C_c)$ after filtering and selection. Our image over-generation pipeline produced 2.45 million candidate image pairs $(I_o^s, I_c^s)$ for these 24.5k caption pairs, of which 17,410 had at least one generated image pair which met our filtering criteria. After selection according to the $\text{CLIP}_{dir}$ metric, a total of 34,820 image-caption pairs remain, comprising our COCO-Counterfactuals dataset[11].

---

[8]We use the implementation from Instruct-Pix2Pix (Brooks et al., 2023).

[9]We use the 5K validation split of the 2017 dataset from `https://cocodataset.org/#download`.

[10]While we use MS-COCO in this study as the source of our original captions, one advantage of our counterfactual generation approach is that the input dataset itself does not require paired image and text data.

[11]While the MS-COCO 5K validation split has 25,014 captions, COCO-Counterfactuals includes only 17,410 of them due to our filtering criteria. Thus, for a fair comparison in our experiments, hereafter we refer to this subset of the 5K validation split including only those 17,410 captions and their paired original images as the MS-COCO dataset.

| Image set | Correct | Incorrect | Neither | Both |
|---|---|---|---|---|
| Generated from original caption | 79.10% | 8.16% | 10.18% | 2.56% |
| Generated from counterfactual caption | 67.27% | 18.43% | 10.74% | 3.56% |
| All images | 73.18% | 13.30% | 10.46% | 3.06% |

Table 1: Human evaluation results for COCO-Counterfactuals

# 4 COCO-Counterfactuals Analysis

This section aims to show that, in addressing the challenges associated with low-resource settings involving multimodal data (see Section 1), our proposed novel data generation pipeline can serve as an efficient and scalable framework to automatically create high quality multimodal counterfactual examples in COCO-Counterfactuals (COCO-CFs). Toward this goal, we first employ human evaluation to analyze COCO-CFs. We then show that COCO-CFs can be used as a challenging dataset for model evaluation on zero-shot image-text retrieval and image-text matching tasks.

## 4.1 Human Evaluation of COCO-Counterfactuals

We employ professional data annotators to conduct a human study on the quality of COCO-CFs. For each of the 34,820 images in COCO-CFs, we have at least one annotator choose whether the corresponding original or counterfactual caption best fits the image. Annotators can also choose "both" if both captions describe the image equally well, or "neither" if neither caption accurately describes the image. 10% of the images are labeled by 3 different individuals to estimate inter-annotator agreement, with the remaining images each labeled by a single annotator (see Appendix B.2 for additional details).

Table 1 provides the percentage of images from COCO-CFs which were matched to their correct caption by the human annotators. We also report the percentage of incorrect matches (i.e., the wrong caption was picked as best describing the image) as well as the percentage of "both" and "neither" labels. Overall, 73% of images were correctly matched to their corresponding caption (see Appendix A.3 for an analysis of incorrect matches). Images generated from the counterfactual caption had a 10% greater incidence of incorrect caption selections than those generated from the original caption. This could be due to the constraints imposed on the counterfactual image by Prompt-to-Prompt (i.e., shared attention weights with the original image), which increases the likelihood that the generated image lacks some of the details in its corresponding caption.

The Fleiss' kappa coefficient for the 10% of images labeled by three annotators was 0.74, indicating strong agreement among the annotators who participated in this study. Among those images which had label disagreement, 47.4% of the labels were correct, 27.3% were incorrect, and 18.6% selected "neither." This suggests that many of the disagreements are associated with images for which the correct caption choice is more ambiguous.

While we employed human annotators to validate the quality of COCO-Counterfactuals for this analysis, our automated counterfactual generation approach does not require the use of human annotators to produce a new dataset. Indeed, our experiments described subsequently in Section 5.1 show that COCO-Counterfactuals which were labeled as incorrect by humans have no negative impact on training data augmentation.

## 4.2 COCO-Counterfactuals for Model Evaluation

Motivated by prior work which has proposed using counterfactuals as challenging test sets in NLP (Kaushik et al., 2019; Gardner et al., 2020), we further investigate whether our COCO-CFs can serve a similar purpose for state-of-the-art multimodal vision-language models such as CLIP, Flava (Singh et al., 2022), BridgeTower (Xu et al., 2022) and ViLT (Kim et al., 2021) for the zero-shot image-text retrieval and image-text matching tasks. We employed the HuggingFace implementations of these model in our experiments (see Appendix A.6 for more detail).

| HuggingFace Pre-trained Models | Evaluated Dataset | Text Retrieval | | | Image Retrieval | | |
|---|---|---|---|---|---|---|---|
| | | R@1 | R@5 | R@10 | R@1 | R@5 | R@10 |
| bridgetower-large-itm-mlm-itc | COCO-CFs | 21.72 (**-51%**) | 46.94 (-35%) | 58.65 (-29%) | 17.93 (-47%) | 38.94 (-35%) | 49.95 (-30%) |
| | human-evaluated COCO-CFs | 26.36 (**-41%**) | 54.1 (-25%) | 66.13 (-20%) | 21.44 (-37%) | 45 (-25%) | 56.39 (-21%) |
| flava-full | COCO-CFs | 21.28 (**-57%**) | 46.64 (-41%) | 58.87 (-34%) | 37.76 (-16%) | 66.15 (-12%) | 75.83 (-10%) |
| | human-evaluated COCO-CFs | 26.1 (**-47%**) | 54.23 (-31%) | 66.83 (-25%) | 43.4 (-3%) | 72.35 (-3%) | 81.44 (-4%) |

Table 2: Image-text retrieval performance on COCO-CFs and human-evaluated COCO-CFs for BridgeTower and Flava models. Largest drops of performance against the baseline are in boldface.

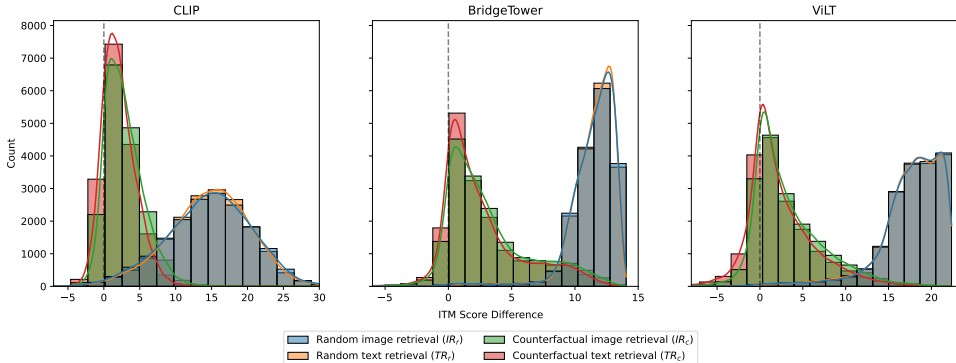

Figure 3: ITM score differences computed for existing multimodal models on COCO-Counterfactuals dataset.

### 4.2.1 Zero-shot Image-text Retrieval

We evaluate the zero-shot image-text retrieval (ITR) performance of pre-trained Flava and BridgeTower models on COCO-CFs as well as *human-evaluated COCO-CFs*, which consists of only image-text pairs that were correctly matched in our human evaluation study (Section 4.1). Since pre-trained CLIP was employed in our counterfactual image generation process (see Section 3.2), it is not suitable for zero-shot ITR setting. Thus, we only report its evaluation in Appendix. A.6 for completeness. For baselines, we evaluate ITR performance of these models on the MS-COCO dataset.

Table 2 reports ITR performance (i.e., Recall at 1, 5, and 10) on COCO-CFs and human-evaluated-COCO-CFs for pre-trained BridgeTower and Flava models. The percentages enclosed within parentheses indicate the change in performance of a model on an evaluated dataset versus the performance of that model on MS-COCO (baseline). We observe that the performance of BridgeTower and Flava decreases significantly (up to $51\%$ and $57\%$, respectively) compared to the baseline's performance on both COCO-CFs and human-evaluated-COCO-CFs. These results demonstrate that COCO-Counterfactuals can serve as a challenging test set for SOTA multimodal vision-language models.

### 4.2.2 Image-text Matching

Typically, during pre-training for image-text matching (ITM), multimodal models learn to differentiate actual image-text pairs from alternative images or captions which are randomly sampled from a dataset. By design, our COCO-CFs have the potential to make this task significantly more challenging by requiring models to also differentiate between minimally-edited image or text candidates. We measure the magnitude of this increased difficulty by comparing the difference in ITM scores between actual image-text pairs and their corresponding counterfactual or randomly sampled alternatives.

Let $(C_o, I_o)$ denote an original image-text pair from MS-COCO, $I_o^s$ denote our synthetically-generated image corresponding to $C_o$, and $(C_c, I_c^s)$ denote our corresponding synthetically-generated counterfactual image-text pair in COCO-Counterfactuals. We further denote $(C_r, I_r)$ as a different original image-text pair randomly sampled from MS-COCO such that $I_o \neq I_r$. For a given pre-trained

multimodal model, we compute the following metrics using its ITM scoring function $\mathcal{G}$:

$$\text{IR}_r = \mathcal{G}(C_r, I_r) - \mathcal{G}(C_r, I_o) \qquad\qquad \text{IR}_c = \mathcal{G}(C_c, I_c^s) - \mathcal{G}(C_c, I_o^s)$$
$$\text{TR}_r = \mathcal{G}(C_r, I_r) - \mathcal{G}(C_o, I_r) \qquad\qquad \text{TR}_c = \mathcal{G}(C_c, I_c^s) - \mathcal{G}(C_o, I_c^s)$$

$\text{IR}_r$ and $\text{TR}_r$ scores can be viewed as measuring the confidence of a model's image or text retrieval (respectively) over two real image-text pairs from MS-COCO. Similarly, $\text{IR}_c$ and $\text{TR}_c$ scores measure image or text retrieval confidence, but using matched image-text pairs from COCO-Counterfactuals dataset. For all metrics, values greater than zero indicate that a model scores the correct image-text pair as more similar than its random or counterfactual alternative. Larger positive values can be viewed as indicating greater confidence in the model's correct discernment between the alternatives.

Figure 3 plots the distribution of these four metrics for three pre-trained multimodal models: CLIP, BridgeTower, and ViLT. All three models exhibit a significant negative distribution shift when presented with counterfactual alternatives rather than random alternatives, demonstrating the increased difficulty of COCO-Counterfactuals for existing models. A significant number of COCO-Counterfactuals are also incorrectly scored (i.e., have ITM score difference less than zero) by all three models. Even in cases where the counterfactual alternatives can be correctly discerned, we posit that the much smaller values of $\text{IR}_c$ and $\text{TR}_c$ may improve the efficiency of training through the increased difficulty of the ITM task.

### 4.2.3 Discussion

When used as a test set, COCO-Counterfactuals by design evaluate the robustness of models to minimal changes in paired image-text data. Table 2 and Figure 3 show that existing models perform significantly worse when evaluated on COCO-Counterfactuals. Additionally, we find that training these same models on COCO-Counterfactuals produces an average relative improvement of 24.3% in image-text retrieval performance on withheld counterfactual examples (see Table 5 of Appendix A.1). These results point to the usefulness our dataset for evaluating and improving the robustness of multimodal models to counterfactual changes.

## 5 COCO-Counterfactuals for Training Data Augmentation

This section aims to evaluate whether COCO-Counterfactuals can serve as an alternative to real data for training data augmentation in low-resource scenarios. We train a fully unfrozen pre-trained CLIP model with its contrastive loss using various combinations of real data from MS-COCO and COCO-CFs datasets (see Appendix B.3 for additional training details). In order to investigate the robustness of models trained on COCO-CFs, we evaluate them on OOD datasets for image-text retrieval and image recognition. For baselines, we report the performance of pre-trained CLIP (i.e., without any additional training) as well as a CLIP model which has been additionally trained using only real data from MS-COCO. We repeat each of our training experiments with 25 different random seeds and report both the mean and standard deviation of performance measured across all random seeds. We also validate the statistical significance of performance improvements obtained by models trained on COCO-CFs using one-tailed t-tests.

### 5.1 Image-text Retrieval

To evaluate OOD performance on the image-text retrieval task that CLIP was trained for, we use the 1K test set of Flickr30k dataset (Young et al., 2014). Table 3 reports the zero-shot performance of the baselines as well as CLIP trained with varying amounts of the original MS-COCO and COCO-CFs datasets. We observe that all CLIP models trained with COCO-Counterfactuals outperform pre-trained CLIP by an average of 5 points, based on the mean performance across text and image retrieval settings. Additionally, our best model trained with 20,894 COCO-Counterfactuals provides statistically significant improvements relative to training only on the real MS-COCO dataset across all settings. We also found that COCO-Counterfactuals improve in-domain performance on the MS-COCO test set, which we detail in Appendix A.5.

To investigate the potential impact of COCO-Counterfactuals which were labeled incorrectly by human annotators, we repeated our training data augmentation experiments using only image-text pairs which were correctly matched in our human evaluation study (Section 4.1). Overall, we found

| Training dataset | $|D_{\text{train}}|$ | % CFs | Text Retrieval | | | Image Retrieval | | | Mean |
|---|---|---|---|---|---|---|---|---|---|
| | | | **R@1** | **R@5** | **R@10** | **R@1** | **R@5** | **R@10** | |
| None (pre-trained CLIP) | 0 | 0% | 67.1 | 89 | 93.8 | 69.4 | 90.6 | 94.9 | 84.13 |
| MS-COCO | 13,928 | 0% | $77.90_{0.4}$ | $93.79_{0.2}$ | $97.11_{0.1}$ | $75.14_{0.4}$ | $93.72_{0.2}$ | $96.69_{0.2}$ | $89.06_{0.1}$ |
| MS-COCO + COCO-CFs | 13,928 | 50% | $76.66_{0.5}$ | $\underline{94.53}_{0.3}$ | $96.84_{0.2}$ | $\underline{75.75}_{0.4}$ | $93.60_{0.2}$ | $\mathbf{96.96}_{0.2}$ | $89.05_{0.1}$ |
| MS-COCO + COCO-CFs | 34,820 | 60% | $\underline{78.28}_{0.4}$ | $\mathbf{94.72}_{0.3}$ | $\mathbf{97.27}_{0.2}$ | $\underline{76.13}_{0.5}$ | $93.85_{0.2}$ | $96.91_{0.2}$ | $\mathbf{\underline{89.53}}_{0.2}$ |
| MS-COCO + COCO-CFs | 41,784 | 67% | $77.75_{0.5}$ | $94.51_{0.3}$ | $97.03_{0.2}$ | $\mathbf{\underline{76.38}}_{0.3}$ | $\mathbf{\underline{94.01}}_{0.2}$ | $96.79_{0.2}$ | $\underline{89.41}_{0.1}$ |

Table 3: Image-text retrieval performance on the OOD Flickr30k 1K test set for pre-trained CLIP and CLIP models trained on varying amounts of data from MS-COCO and COCO-CFs datasets. $|D_{\text{train}}|$ indicates the total number of image-text pairs used for training, while % CFs indicates the percentage of those image-text pairs which were sampled from COCO-CFs. Results report mean over 25 different random seeds, with standard deviation as a subscript. Best results are in boldface. Results which use COCO-CFs are underlined when a one-tailed t-test indicates that their improvement over training only on MS-COCO is statistically significant ($p \leq 0.05$)

| Training dataset | $|D_{\text{train}}|$ | % CFs | CIFAR10 | CIFAR100 | Food101 | Caltech101 | Caltech256 | ImageNet | Mean |
|---|---|---|---|---|---|---|---|---|---|
| None (pre-trained CLIP) | 0 | 0% | 88.8 | 64.17 | **84.17** | 90.32 | 83.43 | 59.25 | 78.36 |
| MS-COCO | 13,928 | 0% | $89.21_{0.3}$ | $63.89_{0.4}$ | $82.67_{0.2}$ | $92.77_{0.1}$ | $85.05_{0.1}$ | $59.55_{0.2}$ | $78.85_{0.2}$ |
| MS-COCO + COCO-CFs | 13,928 | 50% | $\underline{89.45}_{0.3}$ | $\underline{66.67}_{0.4}$ | $\underline{83.13}_{0.2}$ | $92.63_{0.1}$ | $\mathbf{\underline{85.21}}_{0.1}$ | $\mathbf{\underline{59.66}}_{0.2}$ | $\mathbf{\underline{79.46}}_{0.2}$ |
| MS-COCO + COCO-CFs | 34,820 | 60% | $89.16_{0.3}$ | $\mathbf{\underline{66.88}}_{0.4}$ | $82.12_{0.2}$ | $\mathbf{\underline{92.87}}_{0.1}$ | $84.95_{0.2}$ | $59.22_{0.3}$ | $\underline{79.20}_{0.2}$ |
| MS-COCO + COCO-CFs | 41,784 | 67% | $88.51_{0.5}$ | $\underline{65.97}_{0.5}$ | $82.06_{0.2}$ | $92.77_{0.1}$ | $84.59_{0.2}$ | $58.88_{0.2}$ | $78.80_{0.2}$ |

Table 4: Zero-shot classification accuracy of pre-trained CLIP and CLIP models trained on varying amounts of data from MS-COCO and COCO-CFs datasets. All other settings are identical to Table 3.

that excluding these incorrectly-labeled COCO-Counterfactuals from training data augmentation had a negligible impact on performance (see Appendix A.4). This suggests that training data augmentation is robust to noise introduced by synthetic data, and that the 26.82% of incorrectly-labeled COCO-Counterfactuals do not pose an issue for data augmentation applications. While certain use cases which require a high degree of confidence in the accuracy of generated counterfactuals may benefit from the use of human validation, we believe that these results demonstrate how our approach can be used for fully automated training data augmentation without human annotation.

## 5.2 Image Recognition

Despite being trained for image-text retrieval, CLIP has exhibited impressive performance at zero-shot image recognition. Using the same approach as Radford et al. (2021) for the image recognition task (i.e., form a sentence "A photo of a $\{c\}$" for each class label $c$ to obtain image-text matching scores), we evaluate whether CLIP models trained on COCO-Counterfactuals exhibit competitive OOD performance improvement to baselines' performance in this zero-shot classification setting.

Using the same CLIP models trained on varying amounts of MS-COCO and COCO-CFs, Table 4 reports their zero-shot classification accuracy on six image recognition datasets. We observe that training with an approximately 50-50 split of MS-COCO & COCO-CFs provides the best overall performance, offering improvements over pre-trained CLIP (without any additional training) on all datasets except Food101 and outperforming training with only MS-COCO on most datasets (see Appendix A.2 for additional analysis of performance differences).

## 5.3 Discussion

Recent work investigating the suitability of synthetic training data for image recognition tasks has found that synthetic image data is much less efficient than real data, requiring 5x more synthetic training samples to achieve similar performance as models trained on real data (He et al., 2022). In contrast, our results show that training data augmentation with COCO-Counterfactuals is at least as efficient (Table 3) and sometimes more efficient (Table 4) than data augmentation with an identical amount of real data ($|D_{\text{train}}| = 13,928$). This finding suggests that our approach could be particularly valuable in low-resource settings where paired image-text data is scarce.

Consistent with prior work on training data augmentation with NLP counterfactuals (Howard et al., 2022; Joshi and He, 2022), Tables 3 and 4 show that improvements in OOD performance with increasing amounts of counterfactual examples reaches a saturation point, beyond which additional data augmentation does not lead to further improvements. For image-text retrieval on Flickr30k (Table 3), this saturation point is reached with a 40 / 60% mixture of MS-COCO / COCO-Counterfactuals in the training dataset. In contrast, Table 4 shows that the saturation point for the OOD image recognition datasets is reached with a 50 / 50% split based on the mean of the six datasets. These results suggest that the optimal mixture of real examples and synthetically generated counterfactual examples may differ depending on the evaluation task and dataset.

While training data augmentation with COCO-Counterfactuals produces statistically significant performance improvements relative to training with only real data, the overall magnitude of these improvements is limited and varies by evaluation setting. COCO-Counterfactuals produce the largest improvements on zero-shot image recognition tasks, where its overall mean improvement over pre-trained CLIP is twice as large as that achieved by training on an equivalent amount of real data from MS-COCO. However, OOD generalization performance varies by dataset, which further analysis suggests, is related to domain gaps between altered subjects in COCO-Counterfactuals and the domain of the evaluation dataset (see Appendix A.2 for details).

# 6 Conclusion

We proposed an automated data generation methodology for creating counterfactual examples from image-text pairs to address the challenge of low-resource settings involving multimodal data. This approach was used to create COCO-Counterfactuals (COCO-CFs), a high-quality synthetic dataset of paired image-text counterfactuals derived from MS-COCO captions. COCO-CFs are challenging for existing pre-trained multimodal models and significantly increase the difficulty of the zero-shot image-text retrieval and image-text matching tasks. Our experiments demonstrate that augmenting training data with COCO-CFs improves OOD generalization on multiple downstream tasks.

In this work, we focused on the creation of task-agnostic counterfactual examples. A promising direction for future research is the adaptation of our approach to produce task-specific counterfactuals. For example, in the case of image recognition, the counterfactual changes could be limited to a targeted label distribution to produce counterfactual examples more tailored to the end task. Alternatively, task-specific model failures or spurious correlations could be diagnosed and used as a basis for determining which counterfactual changes to consider when creating the counterfactual captions. We believe that such approaches have the potential to produce counterfactuals which are more targeted for improving specific model deficiencies.

Another opportunity for future work is larger-scale automatic generation of counterfactual examples to enable full counterfactual pre-training of multimodal models. Additionally, we believe that extending our image-text counterfactuals to the video domain could be a promising path towards improving video transformers through counterfactual data augmentation.

**Limitations & Ethical Concerns**   Motivated by the desire to produce minimal-edit counterfactuals, we only considered changes to nouns. This is a common strategy for NLP counterfactuals (see Appendix B.1.1 for discussion), but alternative generation strategies such as controlled text decoding (Howard et al., 2022) could be used to enable a larger range of counterfactual changes, in addition to alterations of adjectives or verbs. We leave investigation of these directions to future studies.

Due to a limited compute budget, we only explored generating COCO-Counterfactuals using Stable Diffusion. Additionally, our training data augmentation experiments were limited to a single model (CLIP). It is possible that other text-to-image generation models may exhibit better performance for generating counterfactual image-text data. Additionally, the benefits of counterfactual data augmentation may vary for different multimodal vision-language models.

Despite the impressive recent improvements in text-to-image generation capabilities, models such as Stable Diffusion have well-known limitations that should be considered when utilizing datasets which are derived from them (see Appendix C.5 for a detailed discussion). We do not foresee significant risks of security threats or human rights violations in our work. However, the automated nature of our image generation process may introduce the possibility of our COCO-Counterfactuals dataset containing images that some individuals may consider inappropriate or offensive.

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

# Appendix

## A  Additional Experiments and Analysis

### A.1  COCO-Counterfactuals Improve Model Robustness to Counterfactual Changes

By design, COCO-Counterfactuals may offer greater improvements to the robustness of models to minimal or counterfactual changes in images. Such examples are unlikely to be present in the datasets used previously to evaluate OOD generalization. Therefore, we also evaluate the performance of models on a withheld test set of COCO-Counterfactuals to determine their image-text retrieval capabilities on in-domain counterfactual examples. Specifically, we withhold 30% of the original-counterfactual paired examples in COCO-Counterfactuals for testing and train the pre-trained CLIP, BridgeTower, and Flava models on the remainder, with 56% of the total dataset used for training and 14% used as a development set.

Table 5 compares the performances of CLIP, BridgeTower, and Flava models trained on COCO-Counterfactuals to those trained on an equivalent amount of real examples from MS-COCO and to their pre-trained versions[12]. We observe that training on COCO-Counterfactuals results in a mean improvement of 11.83, 21.55, and 11.47 relative to the pre-trained CLIP, BridgeTower, and Flava models, respectively. This represents an average relative improvement of 24.3% for each model over the performance of its pre-trained version. In addition, the CLIP, BridgeTower, and Flava models that were trained on COCO-Counterfactuals achieve a mean absolute improvement of 6.06, 10.08, and 5.28, respectively, relative to those that were trained on MS-COCO. The greater magnitude of these performance gains relative to our OOD image-text retrieval evaluations (Table 3) suggests that training on COCO-Counterfactuals improves model robustness to counterfactual changes, which are not present in our (non-counterfactual) OOD evaluation datasets.

| Pre-trained Models | Training dataset | Text Retrieval | | | Image Retrieval | | | Mean |
|---|---|---|---|---|---|---|---|---|
| | | R@1 | R@5 | R@10 | R@1 | R@5 | R@10 | |
| CLIP | None (pre-trained CLIP) | 50.96 | 79.33 | 86.45 | 47.89 | 77.19 | 85.73 | 71.26 |
| | MS-COCO | 57.17 | 84.23 | 90.66 | 55.45 | 84.00 | 90.65 | 77.03 |
| | COCO-CFs | 65.03 | 90.26 | 94.99 | 64.09 | 89.52 | 94.62 | 83.09 |
| BridgeTower | None (pre-trained BridgeTower) | 35.26 | 65.31 | 76.73 | 28.77 | 56.63 | 68.46 | 55.19 |
| | MS-COCO | 41.78 | 71.78 | 81.88 | 44.68 | 75.38 | 84.48 | 66.66 |
| | COCO-CFs | 54.37 | 83.08 | 90.53 | 56.63 | 84.48 | 91.36 | 76.74 |
| Flava | None (pre-trained Flava) | 34.40 | 66.63 | 78.02 | 51.55 | 80.64 | 88.24 | 66.58 |
| | MS-COCO | 46.70 | 76.36 | 85.68 | 52.55 | 81.08 | 88.43 | 71.80 |
| | COCO-CFs | 54.39 | 83.35 | 90.27 | 57.97 | 85.11 | 91.38 | 77.08 |

Table 5: Image-text retrieval performance on a withheld COCO-CFs test set.

### A.2  Analysis of Differences in OOD Generalization on Image Recognition Datasets

To better understand the differences in OOD generalization performance across datasets, we measured the frequency in which the altered subjects used to produce COCO-Counterfactuals overlapped with class labels. Specifically, we define the COCO-CFs Label Frequency for each image recognition dataset as the total number of COCO-Counterfactuals in which one or more of the dataset's labels matched one of the two altered subjects used to produce the counterfactual pair.

Table 6 provides the COCO-CFs Label Frequency for each image recognition dataset along with the change in OOD performance relative to pre-trained CLIP after training on various sizes of COCO-CFs (see Appendix B.3.1 for a definition of dataset sizes). We observe that datasets having a higher COCO-CFs Label Frequency generally achieve larger improvements in OOD generalization

---

[12]Note that the image-text retrieval performance of the three pre-trained models (CLIP, BridgeTower, and Flava) on the in-domain COCO-Counterfactuals test set in Table 5 are higher than the respective values on the entire COCO-Counterfactuals dataset provided in Tables 2 and 13. This is expected because the retrieval space of the in-domain COCO-Counterfactuals test set is only 30% of the entire COCO-Counterfactuals dataset.

| IR Dataset | COCO-CFs Label Frequency | COCO-CFs$_{base}$ $\Delta$ | COCO-CFs$_{medium}$ $\Delta$ | COCO-CFs$_{all}$ $\Delta$ |
|---|---|---|---|---|
| CIFAR100 | 3446 | 2.50 | 2.63 | 1.80 |
| Caltech101 | 354 | 2.31 | 2.55 | 2.45 |
| Caltech256 | 744 | 1.78 | 1.52 | 1.16 |
| CIFAR10 | 398 | 0.65 | 0.36 | -0.29 |
| ImageNet | 887 | 0.41 | -0.03 | -0.37 |
| Food101 | 28 | -1.04 | -2.05 | -2.11 |

Table 6: Frequency of class label occurrence in COCO-CFs and absolute change ($\Delta$) in performance relative to pre-trained CLIP after training on various sizes of COCO-CFs

| Error category | % present in sampled COCO-CFs |
|---|---|
| Failure to generate subject/object | 27% |
| Failure to generate fine-grained details | 23% |
| Hyponymy relationship between altered subjects | 15% |
| Human annotation error | 15% |
| Failure to accurately depict spatial relationships | 7% |
| Failure to generate correct number of objects | 6% |
| Both altered subjects are present in the image | 4% |
| Failure to bind attribute | 3% |

Table 7: Image-text retrieval performance on the in-domain COCO-CFs test set.

performance. The Pearson correlation coefficient between COCO-CFs Label Frequency and the 18 performance change measurements in Table 6 is 0.522 with a p-value of 0.026, indicating statistically significant positive correlation.

These results suggest that a major contributor to the variation in OOD generalization performance across datasets is the overlap between the evaluation dataset domain and the set of subjects which are altered in COCO-Counterfactuals. Food101, the only dataset which saw no improvement in performance on our best-performing COCO-CFs training dataset, had only 28 cases of overlap between its label set and the subject alterations in COCO-CFs. In contrast, the greatest performance improvements were achieved on CIFAR100, for which 3446 COCO-CFs had subject alterations matching at least one label from the dataset. These findings point to the potential usefulness of targeting counterfactual changes for task-specific datasets.

## A.3 Analysis of Errors in COCO-Counterfactuals Identified by Human Annotators

In this section, we analyze errors in COCO-Counterfactuals using the labels assigned by human annotators (Section 4.1). Specifically, we consider an error to be any image-text pair from the COCO-Counterfactuals dataset for which the human annotator did not select the correct caption for the corresponding image.

### A.3.1 Manual Categorization of Errors

To investigate potential failure cases in our counterfactual generation approach, we randomly sampled and categorized 100 image-text pairs which were identified as errors by the human annotators. Table 7 provides the percentage of sampled COCO-Counterfactuals which were assigned to various error categories. Additionally, Tables 8 and 9 provide examples of counterfactual pairs which were assigned to the top-six most frequent error categories.

We found that 66% of the sampled errors can be attributed to known limitations of existing text-to-image diffusion models (Chefer et al., 2023; Samuel et al., 2023; Cho et al., 2022), which include the categories for failure to generate a subject or object (e.g., Table 8, row 1), failure to generate fine-grained details (e.g., Table 8, row 2), failure to accurately depict spatial relationships (e.g., Table 9, row 2), failure to generate the correct number of objects described in the prompt (e.g., Table 9, row 3), and failure to bind attributes such as color.

| Original | Counterfactual |
|:--:|:--:|

Failure to generate subject/object

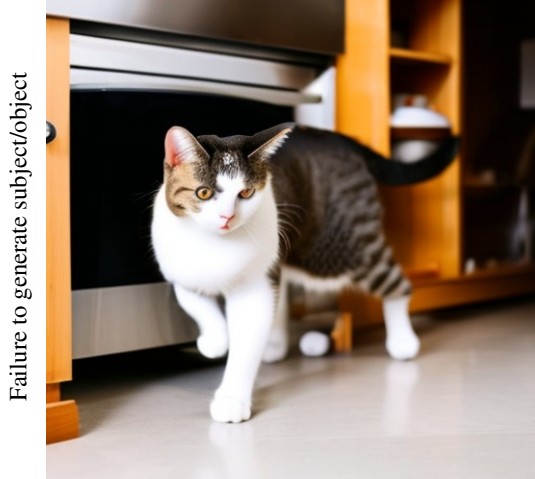 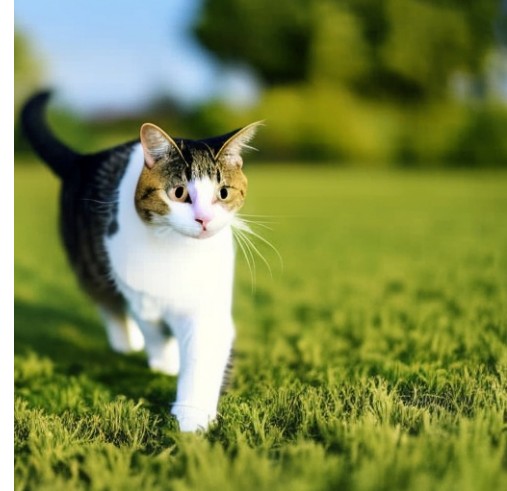

*A cat walking through a **kitchen** by a eating tray*     *A cat walking through a **field** by a eating tray.*

Failure to generate fine-grained details

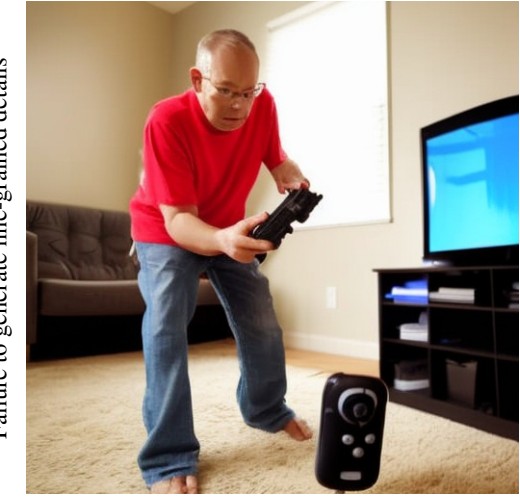 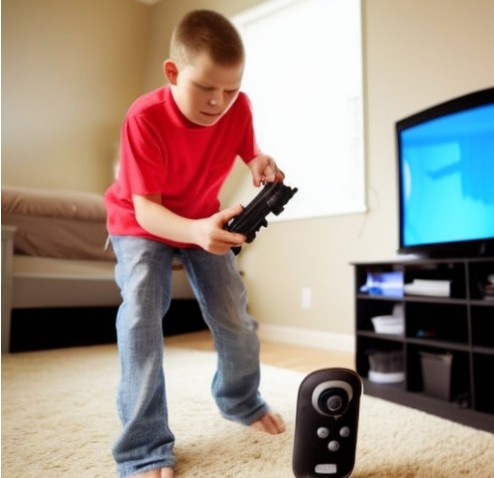

*A **man** playing Wii in a dirty room*     *A **kid** playing Wii in a dirty room*

Hyponymy relationship between altered subjects

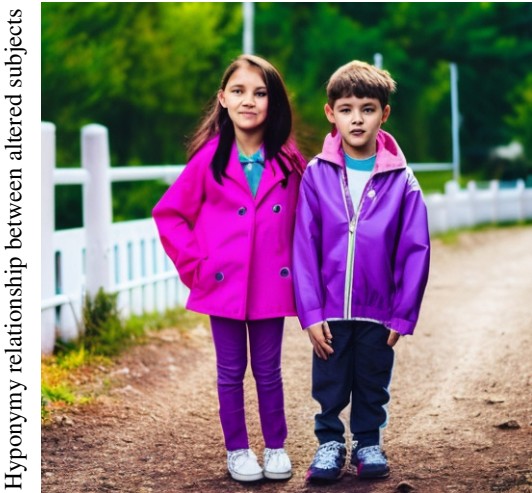 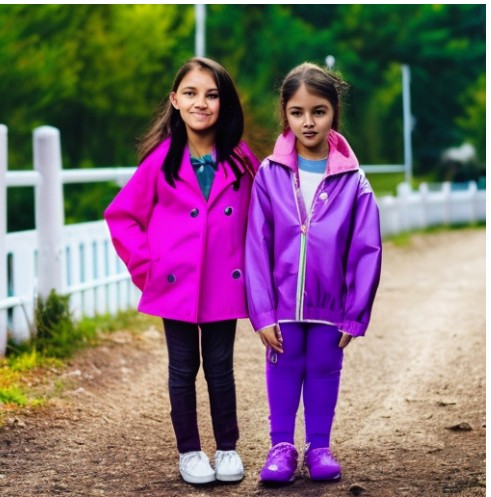

*Two **kids** in pink and purple jackets standing by a fence*     *Two **girls** in pink and purple jackets standing by a fence*

Table 8: Examples of failure cases identified by manual error analysis

|  | **Original** | **Counterfactual** |
|---|---|---|

Human annotation error

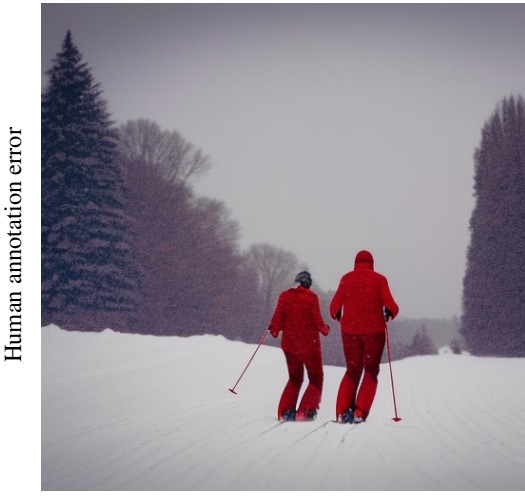 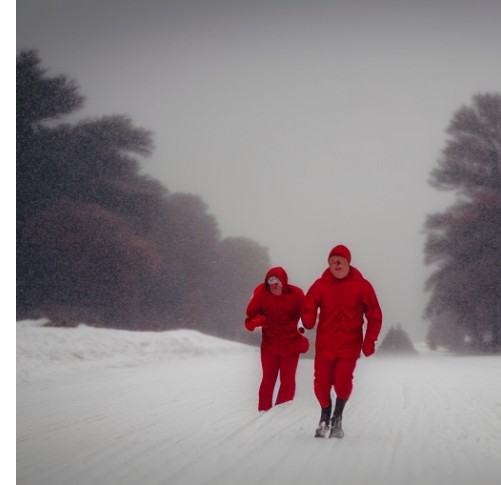

*Two people dressed in red **skiing** across a snowy landscape*

*Two people dressed in red **race** across a snowy landscape*

Failure to accurately depict spatial relationships

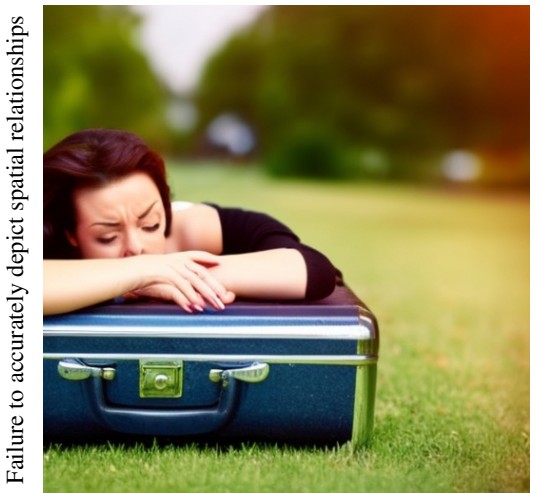 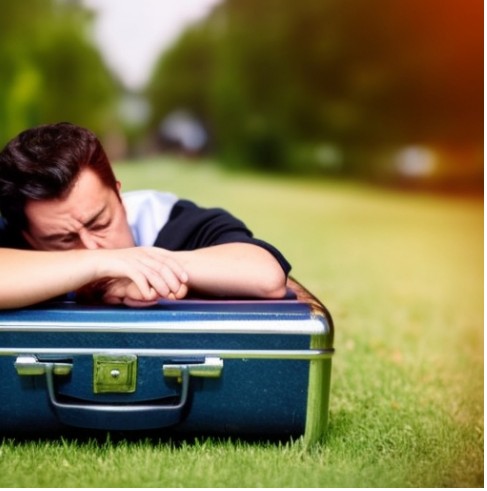

*A **woman** lies on the ground under a suitcase.*

*A **man** lies on the ground under a suitcase.*

Failure to generate correct number of objects

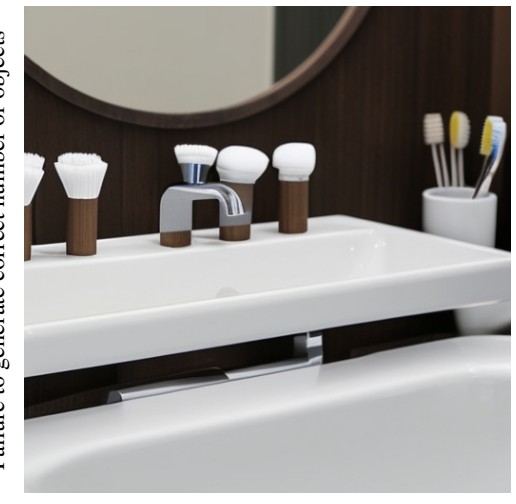 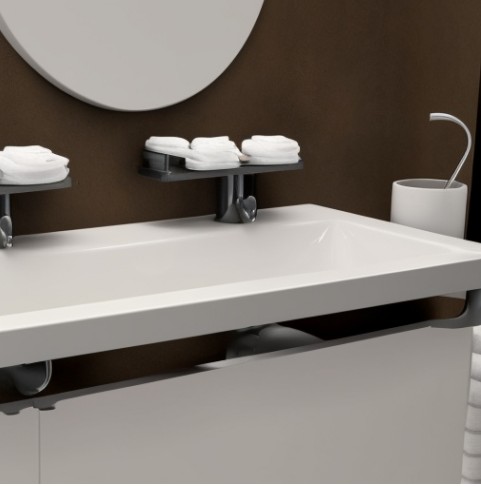

*A bathroom sink with two **toothbrush** holders on it*

*A bathroom sink with two **cup** holders on it*

Table 9: Additional examples of failure cases identified by manual error analysis

| Altered Subjects | Count | Altered Subjects | Count | Altered Subjects | Count |
|---|---|---|---|---|---|
| woman → girl | 126 | man → boy | 125 | people → men | 116 |
| person → man | 93 | person → woman | 42 | person → boy | 37 |
| couple → group | 36 | people → guy | 35 | people → kid | 33 |
| person → girl | 33 | girl → woman | 32 | man → woman | 30 |
| men → people | 29 | people → student | 27 | woman → man | 24 |
| man → person | 24 | building → house | 23 | men → boy | 21 |
| women → girl | 21 | boy → man | 21 | | |

Table 10: Frequency of altered subjects which appeared at least 20 times in errors identified by human annotators

In many cases, these failures do not negatively impact the depiction of the counterfactual change in the two images because the inaccuracies pertain to details other than the altered subjects. For example, the first row of Table 8 shows the counterfactual pair associated with an image which was categorized as a failure to generate a subject/object; in this case, the altered subjects (kitchen → field) are depicted correctly, but both images lack the *eating tray* described in the prompt. Similarly, the counterfactual pair shown in the second row of Table 8 lacks fine-grained details in the prompt (e.g., *dirty* room), but still depicts the altered subjects correctly (man → kid).

We found that 15% of the sampled errors could be attributed to a hyponymy relationship between the altered subjects which caused both captions to be equally valid for a given image. For example, the third row of Table 8 shows a counterfactual pair where the counterfactual image was incorrectly labeled by the human annotator because both captions were valid descriptions of the image (i.e., *girls* can also be referred to as *kids*). Nevertheless, this example is still a valid counterfactual pair considering that the counterfactual caption does not accurately describe the original image and is more descriptive of the counterfactual image than the original caption.

An additional 15% of the sampled errors appeared to be valid image-text pairs without any significant deficiencies. We therefore concluded that such cases were human annotation errors (see Table 9 row 1 for an example). Finally, 4% of the sampled images had equally valid caption choices because both of the altered subjects appeared in the image that was annotated.

The results of this error analysis suggest that the quality of counterfactuals produced by our approach may improve as the capabilities of text-to-image diffusion models advance. New models which overcome known limitations of existing models could be used as a substitute for Stable Diffusion in our approach to produce higher-quality counterfactuals. Additionally, errors associated with hyponymy relationships could be addressed in future work through a refinement of our subject alteration process. For example, ontologies could be used to avoid noun substitutions where it can be determined that a hyponymy relationship exists between the noun candidates. Finally, additional constraints on the image generation process could be explored to prevent both altered subjects from appearing in the same image.

### A.3.2 Taxonomic Analysis of Errors

To better understand the relationship between the altered subjects in our counterfactuals and potential failure cases, we conducted a taxonomic analysis of the altered subjects which occurred most frequently among errors identified by human annotators. Table 10 provides the frequency of altered subject pairs which occurred at least 20 times in the error cases identified by human annotators. Interestingly, we observe that 19 of these 20 most frequent altered subject pairs belong to the *human* taxonomy.

We further analyzed this *human* taxonomy in COCO-Counterfactuals by constructing a list of human-related words, which consists of 'girl', 'boy', 'man', 'men', 'woman', 'guy', 'kid', 'person', 'people', 'child', 'children', 'couple', 'group', and 'lady'. An image-text pair is said to be related to this human taxonomy if the altered subject of its caption belong to this list. We find that there are 4117 image-text pairs in COCO-Counterfactuals that are related to the human taxonomy, among which 1864 were identified as errors by human annotators. The corresponding error rate for altered subjects related to the human taxonomy is 44.3%, which indicates that generating counterfactual pairs involving human altered subjects is more challenging for our approach. This suggests that a

| Training dataset | $|D_{\text{train}}|$ | $|D_{\text{train}}^{\text{CF}}|$ | Text Retrieval | | | Image Retrieval | | | Mean |
|---|---|---|---|---|---|---|---|---|---|
| | | | R@1 | R@5 | R@10 | R@1 | R@5 | R@10 | |
| MS-COCO + COCO-CFs | 34,313 | 20,385 | 75.91 | 93.95 | 96.90 | 77.66 | 94.51 | 97.20 | 89.36 |

Table 11: Mean image-text retrieval performance on the OOD Flickr30k test set using only COCO-Counterfactuals which were correctly labeled by humans, measured across 25 different random seeds.

| Training dataset | $|D_{\text{train}}|$ | $|D_{\text{train}}^{\text{CF}}|$ | Text Retrieval | | | Image Retrieval | | | Mean |
|---|---|---|---|---|---|---|---|---|---|
| | | | R@1 | R@5 | R@10 | R@1 | R@5 | R@10 | |
| None (pre-trained CLIP) | 0 | 0 | 50.12 | 75.04 | 83.6 | 30.73 | 56.28 | 67.18 | 60.49 |
| MS-COCO | 13,928 | 0 | $57.33_{0.3}$ | $81.28_{0.2}$ | $88.71_{0.2}$ | $41.13_{0.1}$ | $68.46_{0.1}$ | $78.45_{0.1}$ | $69.23_{0.1}$ |
| MS-COCO + COCO-CFs | 13,928 | 6,939 | $56.91_{0.3}$ | $80.70_{0.2}$ | $87.82_{0.2}$ | $39.92_{0.1}$ | $67.01_{0.1}$ | $77.15_{0.1}$ | $68.25_{0.1}$ |
| MS-COCO + COCO-CFs | 34,820 | 20,894 | $\mathbf{58.06}_{0.3}$ | $\mathbf{81.39}_{0.2}$ | $\mathbf{88.91}_{0.2}$ | $\underline{41.63}_{0.2}$ | $\underline{68.64}_{0.1}$ | $\underline{78.85}_{0.1}$ | $\underline{69.58}_{0.1}$ |
| MS-COCO + COCO-CFs | 41,784 | 27,853 | $\underline{58.02}_{0.3}$ | $\underline{81.39}_{0.2}$ | $88.78_{0.2}$ | $\mathbf{41.82}_{0.1}$ | $\mathbf{68.79}_{0.1}$ | $\mathbf{78.89}_{0.1}$ | $\mathbf{69.62}_{0.1}$ |

Table 12: Image-text retrieval performance on the in-domain MS-COCO test set. All other settings are identical to Table 3.

promising direction for future work is the exploration of improvements to the generation of images involving human subjects.

### A.4 Training Data Augmentation with Only Correctly-annotated COCO-Counterfactuals

We investigate the potential impact of COCO-Counterfactuals which were incorrectly labeled by humans on training data augmentation. Table 11 provides the OOD image-text retrieval performance in this setting, where COCO-Counterfactuals were filtered to only include those which were correctly labeled by the human annotators. Overall we find similar performance as our previous experiments using the full COCO-Counterfactuals dataset (Table 3), suggesting that filtering our synthetic data using human evaluations is not necessary for data augmentation applications.

### A.5 COCO-Counterfactuals Improve In-domain Performance

We evaluate the same models trained with counterfactual data augmentation described in Section 5 on the MS-COCO test set. The results of this in-domain evaluation are provided in Table 12. Similar to the OOD image-text retrieval setting, we find that data augmentation with $20,892$ COCO-Counterfactuals provides statistically significant performance improvements relative to training without counterfactual data augmentations. Notably, previous work has observed that counterfactual data augmentation can degrade performance on withheld in-domain test sets (Wang and Culotta, 2021; Howard et al., 2022), whereas data augmentation with our COCO-Counterfactuals actually increases in-domain performance on MS-COCO.

### A.6 COCO-Counterfactuals for Model Evaluation Experiments

We further investigate whether our COCO-Counterfactuals (COCO-CFs) can serve as a challenging test set for state-of-the-art multimodal vision-language models such as CLIP, Flava (Singh et al., 2022), BridgeTower (Xu et al., 2022) and ViLT (Kim et al., 2021) for the zero-shot image-text retrieval and image-text matching tasks. We employed the following HuggingFace implementations of these models via the transformers library:

- **CLIP**: We used the pre-trained model clip-vit-base-patch32
- **Flava**: We used the pre-trained model flava-full
- **BridgeTower**: We used the pre-trained model bridgetower-large-itm-mlm-itc
- **ViLT**: We used the pre-trained model vilt-b32-finetuned-coco

**Zero-shot Image-text Retrieval**. In Section 4, we evaluated the zero-shot image-text retrieval (ITR) performance of pre-trained Flava and BridgeTower models on COCO-CFs and *human-evaluated*

| HuggingFace Pre-trained Models | Evaluated Dataset | Text Retrieval | | | Image Retrieval | | |
|---|---|---|---|---|---|---|---|
| | | R@1 | R@5 | R@10 | R@1 | R@5 | R@10 |
| Clip | COCO-CFs | 37.65 (**-21%**) | 64.89 (-9%) | 74.57 (-7%) | 34.98 (+5%) | 62.29 (+7%) | 72.43 (+4%) |
| | human-evaluated-COCO-CFs | 43.25 (**-9%**) | 70.4 (-2%) | 79.37 (-1%) | 40.14 (+21%) | 67.86 (+16%) | 77.66 (+11%) |

Table 13: Image-text retrieval performance on COCO-CFs and human-evaluated COCO-CFs for CLIP model. Largest drops of performance against the baseline are in boldface.

*COCO-CFs* that consists of only image-text pairs that were correctly matched in human evaluation in Section 4.1. Since a pre-trained CLIP model was employed in our counterfactual image generation process (see Section 3.2), CLIP models are not suitable for the zero-shot ITR evaluation. Hence, we only report evaluation of pre-trained CLIP model for ITR task here for completeness.

Table 13 reports ITR performance (i.e., Recall at 1, 5, and 10) on COCO-CFs and human-evaluated-COCO-CFs for the pre-trained CLIP model. Similar to Table 2, the percentages enclosed within parentheses indicate the change in performance of the CLIP model on an evaluated dataset versus the performance of that model on MS-COCO (baseline).

We observe that on both COCO-CFs and human-evaluated-COCO-CFs datasets, while the performance of the pre-trained CLIP model degrades marginally on Text Retrieval task, its performance increases for Image Retrieval task. We attribute this to potential data contamination due to how we employed a pre-trained CLIP model in our counterfactual image generation process (see Section 3.2). As a result, COCO-Counterfactuals includes image-text pairs for which CLIP achieves high image-text retrieval performance.

# B  Dataset and Experiment Details

## B.1  Hyper-parameter Selection and Models Used to Generate COCO-Counterfactuals

In this section, we will detail hyper-parameters and pre-trained models used to our generate COCO-Counterfactuals dataset.

### B.1.1  Creating Counterfactual Captions

Given an original caption from the MS-COCO dataset, we use Natural Language Toolkit (**NLTK**) (Bird et al., 2009) modules:

- *punkt* for sentence tokenizer, and
- *averaged_perceptron_tagger* for part-of-speech (POS) tagger

to identify all nouns as candidate words for substitution.

For each of the identified nouns, we create 10 candidate counterfactual captions by replacing only one noun with the [MASK] token and retrieving the top-10 most probable replacements via masked language modeling (MLM). For MLM, we used the pre-trained model *roberta-base* (Liu et al., 2019) implemented in the library *transformers* (Wolf et al., 2019)

A motivation for our use of noun substitutions is the desire to produce minimal-edit counterfactuals. This is a common strategy for NLP counterfactuals (Kaushik et al., 2019; Wang and Culotta, 2021; Yang et al., 2021) because the high degree of similarity between the original and counterfactual text preserves spurious correlations that models might rely on for discernment. Furthermore, our use of perplexity filtering mitigates the potential for such word substitutions to produce unrealistic counterfactual captions.

In order to measure similarity between each candidate counterfactual caption and an original caption, we used the pre-trained model all-MiniLM-L6-v2, which is implemented within the library *sentence-transformers* (Reimers and Gurevych, 2019).

Among generated candidate counterfactual captions, we kept only those candidates which have a sentence similarity within the range $(0.8, 0.91)$. We selected this similarity range heuristically, observing that it produced best results after extensive experimentation.

Finally, we employed the pre-trained model gpt2-large, a *GPT-2* (Radford et al., 2018) model implemented in the transformers library, to score the perplexity and choose the candidate having the lowest perplexity as our counterfactual caption.

### B.1.2 Generating Counterfactual Images

After creating a counterfactual caption, our next task is to generate synthetic images from the corresponding original caption and counterfactual caption, respectively. In order to do so, we have adopted an implementation from Instruct-Pix2Pix (Brooks et al., 2023) in which all hyper-parameters are set to their default values.

Specifically, we over-generate 100 image pairs with Prompt-to-Prompt by randomly sampling values of the parameter $p \sim U(0.1, 0.9)$ (i.e., parameter $p$ indicates the portion of denoising for which to fix self attention maps). The resulting 100 image pairs are filtered using CLIP (Radford et al., 2021) to ensure:

> *i.* a minimum cosine similarity of 0.2 between the encoding of each caption and its corresponding generated image, and
>
> *ii.* a minimum cosine similarity of 0.7 between the encoding of the two respective images in each generated image pair.

From remaining image pairs, the best image pair is chosen such that it has the highest directional similarity $CLIP_{dir}$ score. Selecting images with the highest $CLIP_{dir}$ improves the overall quality of our generated counterfactuals via greater consistency between the alterations made in both modalities.

### B.2 Human Annotation Study

Professional annotation services for our human study were provided by Mindy Support. The total cost of this study was $1068.59 for 218 annotation hours. The instructions provided to annotators are depicted in Figure 4. We are unable to provide the hourly wages paid to workers as this is considered proprietary information by Mindy Support. However, the following statement was provided by the vendor regarding compensation:

"We prioritize compliance with all standards of local and international legislation, ensuring fair treatment and equal opportunities for individuals of various backgrounds, ages, and other characteristics. We are committed to upholding the principles of fair wages, non-discrimination, and labor standards, including the prohibition of child labor. As an organization, we strictly adhere to legal requirements and strive to create an inclusive and ethical working environment for all. Rest assured that our compensation rates reflect market demands and provide fair remuneration for the work performed by our participants. We remain dedicated to abiding by all labor regulations and social and economic standards."

### B.3 Training Data Augmentation Experiments

In this section, we detail how we constructed our training datasets and how we finetuned the pre-trained CLIP model for experiments described in Section 5.

### B.3.1 Training Dataset Preparation

Our training data augmentation experiments utilize various combinations of the MS-COCO validation set and our COCO-Counterfactuals dataset. For simplicity, a caption-image pair is referred to as a *sample*. We define a *counterfactual sample* as following. Given a sample $(C, I)$ (i.e., caption $C$ and image $I$) from our COCO-Counterfactuals dataset, a sample $(C', I')$ from COCO-Counterfactuals dataset is called a counterfactual sample of $(C, I)$ iff $C'$ and $C$ are counterfactual captions of each other. By this definition, COCO-Counterfactuals dataset includes 34,820 samples that correspond to 17,410 paired counterfactual samples.

For experiments in Section 5, we have prepared the following 4 datasets:

Instructions:

Select the caption which best describes the image. In cases where both captions are valid for the image, please try to pick the one which is more descriptive or detailed. If both captions are valid and describe the image equally well, select "Both". If neither of the captions accurately describe the image, select "Neither".

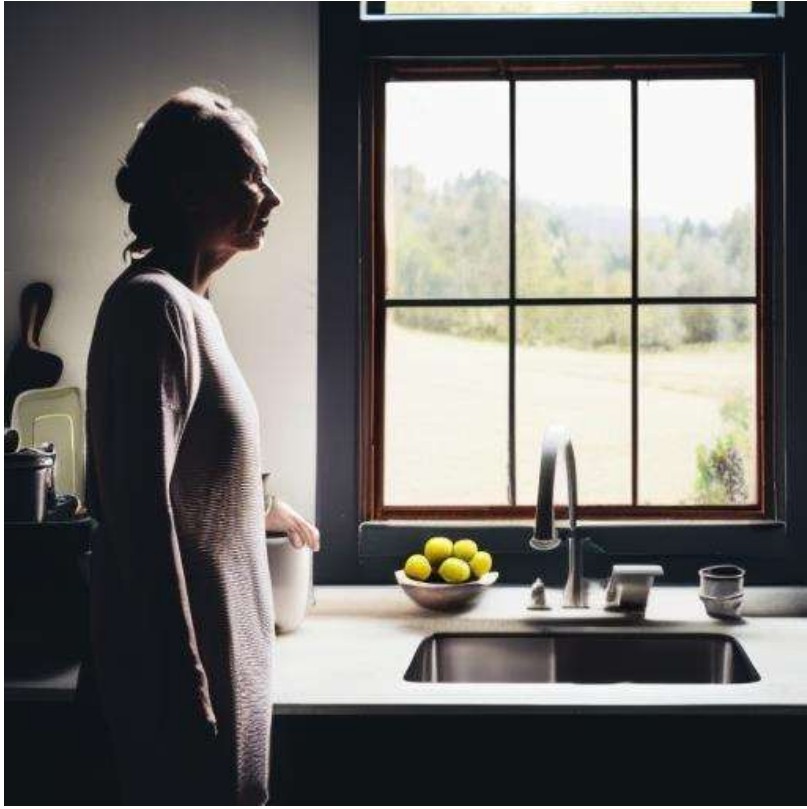

- A woman standing in a kitchen by a window
- A man standing in a kitchen by a window
- Both
- Neither

Figure 4: Instructions provided to data annotators

*(a.)* **MS-COCO** dataset. This is a subset of the 5K validation split of the 2017 MS-COCO dataset[13], achieved by filtering out all samples with captions which are not included in our COCO-Counterfactuals. This results in a dataset (referred to as the MS-COCO dataset used in experiments in Section 5) of 17,410 captions and their paired original images.

*(b.)* **[MS-COCO + COCO-CFs ]$_{base}$** dataset. This dataset is a combination of:

- 50% random sampling (i.e., 8,705 caption-image pairs) of the MS-COCO dataset constructed in *(a.)*.
- 25% random sampling of paired counterfactual samples from our COCO-Counterfactuals dataset. This results in a total of 4,353 pairs of samples with their corresponding counterfactuals, for a total of 8,706 caption-image samples from our COCO-Counterfactuals dataset.

---

[13]https://cocodataset.org/#download

Overall, the [MS-COCO + COCO-CFs ]$_{base}$ dataset consists of 17,411 captions and their paired original images, which is approximately equal in size to the MS-COCO dataset constructed in *(a.)*

*(c.)* **[MS-COCO + COCO-CFs ]$_{\textbf{medium}}$** dataset. This dataset is a combination of:

- all samples (i.e., 17,410 caption-image pairs) from the MS-COCO dataset constructed in *(a.)*.
- 75% random sampling (i.e., 26,115 caption-image pairs) from our COCO-Counterfactuals dataset.

Overall, dataset [MS-COCO + COCO-CFs ]$_{medium}$ consists of 43,525 captions and their paired original images.

*(d.)* **[MS-COCO + COCO-CFs ]$_{\textbf{all}}$** dataset. This dataset is a combination of:

- all samples (i.e., 17,410 caption-image pairs) from the MS-COCO dataset constructed in *(a.)*.
- all samples (i.e., 34,820 caption-image pairs) from our COCO-Counterfactuals dataset.

Overall, dataset [MS-COCO + COCO-CFs ]$_{all}$ consists of 52,230 captions and their paired original images.

Each of the datasets described above is split into a training set (80%) and a validation set (20%). In each experiment, the validation set is used to pick the best model checkpoint at the conclusion of training. Tables 3, 4, and 12 report experimental results for models trained using the train split of these four datasets. $|D_{\text{train}}|$ indicates the total number of samples (i.e., image-text pairs) included in the respective training set, while $|D_{\text{train}}^{\text{CF}}|$ indicates how many of those image-text pairs were sampled from the COCO-Counterfactuals dataset.

### B.3.2 Finetuning CLIP with Data Augmentation

We use each of the four training sets constructed in Section B.3.1 to finetune the CLIP model *clip-vit-base-patch32*. We adopted a publicly-available finetuning script provided by HuggingFace[14].

We repeat each of our training experiments with 25 different *seeds* and *data_seed* from the ranges [107, 131] and [108, 132], respectively. In each experiment, we use a learning rate to 5e-7, weight decay of 0.001, training batch size of 128, and evaluation batch size of 128.

### B.4 Compute Infrastructure Used In this Study

Our experiments were conducted using an Intel AI supercomputing cluster comprised of Intel Xeon processors and 512 Intel Gaudi®AI accelerators, as well as an internal Slurm linux cluster with Nvidia RTX 3090 GPUs. Our dataset generation pipeline was parallelized across this compute infrastructure and took approximately 3 days to complete. Our training data augmentation experiments varied in running time depending upon the size of the dataset, ranging between 2 to 10 hours.

### B.5 License Information of Assets Employed in This Study

- **NLTK** is open source software distributed under the terms of the Apache License Version 2.0.
- *Transformers* is released under the Apache License Version 2.0 and is available on GitHub at `https://github.com/huggingface/transformers`.
- Pre-trained model Roberta-base is released under the MIT License.
- Library *sentence-transformers* is licensed under the Apache License Version 2.0 and is available on GitHub at https://github.com/UKPLab/sentence-transformers.
- Pre-trained model *all-MiniLM-L6-v2* is licensed under the Apache License Version 2.0.
- Pre-trained *gpt2-large* model is license under the MIT License.

---

[14]The finetuning script can be accessed at `https://github.com/huggingface/transformers/blob/main/examples/pytorch/contrastive-image-text/run_clip.py`.

- *Instruct-Pix2Pix* is licensed under the MIT License and is available on GitHub at `https://github.com/timothybrooks/instruct-pix2pix`.
- Instruct-Pix2Pix further employs stable-diffusion-v1-5 that is released under CreativeML-Open-RAIL-M License.
- For the *MS-COCO* dataset:
  - The annotations in the dataset are released under the Creative Commons Attribution 4.0 License.
  - The use of the images in the dataset must abide by the Flickr Terms of Use.
- Pre-trained model *clip-vit-base-patch32* is licensed under the MIT License.
- Pre-trained model *flava-full* is licensed under the 3-Clause BSD License.
- Pre-trained model *BridgeTower large-itm-mlm-itc* is released under the MIT License.
- Pre-trained *vilt-b32-finetuned-coco* model is license under the Apache License Version 2.0.

## C   Datasheet for Dataset

### C.1   Motivation

**For what purpose was this dataset created?** This dataset was created for the purpose of exploring the relevancy of counterfactual examples for multimodal vision-language models. Specifically, our aim was to create a dataset which can serve both as a challenging evaluation dataset for existing models and as a resource for training data augmentation to improve multimodal models on downstream tasks. For additional discussion of our motivation and the intuition behind counterfactual examples, see Section 1.

**Who created the dataset (e.g., which team, research group) and on behalf of which entity (e.g., company, institution, organization)?** The dataset was created by the authors of this paper who are affiliated with Intel Labs, a research and development organization within Intel Corporation.

**Who funded the creation of the dataset?** The creation of this dataset was funded by Intel Corporation.

### C.2   Composition

**What do the instances that comprise the dataset represent (e.g., documents, photos, people, countries)?** The instances represent synthetically-generated images and accompanying text captions. The images depict a variety of different everyday scenarios.

**How many instances are there in total (of each type, if appropriate)?** COCO-Counterfactuals contains a total of 34,820 image-caption pairs.

**Does the dataset contain all possible instances or is it a sample (not necessarily random) of instances from a larger set?** Yes, it contains all possible instances per our filtering criteria.

**What data does each instance consist of?** Each instance consists of a synthetically-generated image and an accompanying text caption.

**Is there a label or target associated with each instance?** No

**Is any information missing from individual instances?** No

**Are relationships between individual instances made explicit (e.g., users' movie ratings, social network links)?** Yes, instances which correspond to a single counterfactual pair are annotated as such in our dataset. Otherwise, there are no other relationships between individual instances.

**Are there recommended data splits (e.g., training, development/validation, testing)?** No

**Are there any errors, sources of noise, or redundancies in the dataset?** The automated methodology used to generate COCO-Counterfactuals introduces the possibility of noise and errors in the dataset. See Section **??** for additional discussion.

**Is the dataset self-contained, or does it link to or otherwise rely on external resources (e.g., websites, tweets, other datasets)?** Yes

**Does the dataset contain data that might be considered confidential (e.g., data that is protected by legal privilege or by doctor–patient confidentiality, data that includes the content of individuals' non-public communications)?** No

**Does the dataset contain data that, if viewed directly, might be offensive, insulting, threatening, or might otherwise cause anxiety?** Yes, the dataset may contain offensive material due to the manner in which it was automatically constructed. See Section **??** for additional discussion.

**Does the dataset identify any subpopulations (e.g., by age, gender)?** No

**Is it possible to identify individuals (i.e., one or more natural persons), either directly or indirectly (i.e., in combination with other data) from the dataset?** No

**Does the dataset contain data that might be considered sensitive in any way (e.g., data that reveals race or ethnic origins, sexual orientations, religious beliefs, political opinions or union memberships, or locations; financial or health data; biometric or genetic data; forms of government identification, such as social security numbers; criminal history)?** No

### C.3 Collection Process

**How was the data associated with each instance acquired?** The data associated with each instance was acquired via our data generation methodology (see Section 3 for a detailed description).

**What mechanisms or procedures were used to collect the data (e.g., hardware apparatuses or sensors, manual human curation, software programs, software APIs)?** Please see Section 3 for a complete description of our data generation methodology.

**If the dataset is a sample from a larger set, what was the sampling strategy (e.g., deterministic, probabilistic with specific sampling probabilities)?** Not applicable

**Who was involved in the data collection process (e.g., students, crowdworkers, contractors) and how were they compensated (e.g., how much were crowdworkers paid)?** The COCO-Counterfactuals dataset was collected automatically, as detailed in Section 3. Human evaluation of COCO-Counterfactuals involved paid professional annotators employed by Mindy Support (see Appendix B.2 for details).

**Over what timeframe was the data collected?** The data was generated and evaluated over the course of approximately three months.

**Were any ethical review processes conducted (e.g., by an institutional review board)?** No, institutional review was not required.

**Did you collect the data from the individuals in question directly, or obtain it via third parties or other sources (e.g., websites)?** No, the dataset was generated automatically and was not collected directly from individuals.

**Were the individuals in question notified about the data collection?** Not applicable

**Did the individuals in question consent to the collection and use of their data?** Not applicable

**If consent was obtained, were the consenting individuals provided with a mechanism to revoke their consent in the future or for certain uses?** Not applicable

**Has an analysis of the potential impact of the dataset and its use on data subjects (e.g., a data protection impact analysis) been conducted?** No, not applicable

### C.4 Preprocessing/cleaning/labeling

**Was any preprocessing/cleaning/labeling of the data done (e.g., discretization or bucketing, tokenization, part-of-speech tagging, SIFT feature extraction, removal of instances, processing of missing values)?** Yes, we apply extensive filtering to various stages of our data generation pipeline in order to improve the quality of the dataset. See Section 3 for a complete description of these methods.

**Was the "raw" data saved in addition to the preprocessed/cleaned/labeled data (e.g., to support unanticipated future uses)?** No. However, due to how our dataset is automatically constructed, raw data can be reproduced by running our code.

**Is the software that was used to preprocess/clean/label the data available?** Yes, we will make our code publicly available upon publication.

### C.5 Uses

**Has the dataset been used for any tasks already?** Yes, we applied COCO-Counterfactuals to the task of model evaluation in Section 4 and to the task of training data augmentation in Section 5.

**Is there a repository that links to any or all papers or systems that use the dataset?** Our GitHub repository will contain links to papers and systems used by our data generation methodology. Additionally, this paper contains references to all such papers and systems that we utilized.

**What (other) tasks could the dataset be used for?** COCO-Counterfactuals is broadly applicable to tasks which require multimodal inputs consisting of images with paired text. One potential use case not explored during this study is large-scale pre-trianing of multimodal models, which could be improved through counterfactual data augmentation.

**Is there anything about the composition of the dataset or the way it was collected and pre-processed/cleaned/labeled that might impact future uses?** Due to the way in which COCO-Counterfactuals was generated automatically, it may contain errors, offensive material, or biases which are present in the models employed by our pipeline.

We used Stable Diffusion to collect image data, which has well-known limitations that should be considered when utilizing datasets which are derived from them. These limitations include unrealistic depictions of hands, palms, and other fine-grained objects (Samuel et al., 2023); failures to generate one or more of the subjects in a prompt and correctly bind attributes such as color (Chefer et al., 2023); difficulties with object counting and spatial relationship understanding (Cho et al., 2022); and challenges associated with the composition of concepts (Liu et al., 2022). While our experiments suggest that COCO-Counterfactuals is relatively robust to generation failures when used for training data augmentation, future applications of our methodology should consider the risks associated with these limitations relative to the intended use of the generated dataset.

Stable diffusion and other text-to-image diffusion models have been shown to exhibit biases associated with race and gender, including over-representation of masculinity and whiteness (Luccioni et al., 2023); racial and gender disparities in depictions of certain occupations (Bianchi et al., 2023); and preferences for certain genders or skin tones (Cho et al., 2022). Consequently, models trained on COCO-Counterfactuals may learn similar social biases as those expressed in synthetic images generated by Stable Diffusion. While some recent work has investigated approaches for mitigating biases in diffusion models, further investigation is needed into the de-biasing of datasets on which these models are trained in order to fully eliminate them (Schramowski et al., 2023).

Users of the dataset should carefully consider how these limitations may impact their potential use case.

**Are there tasks for which the dataset should not be used?** The dataset should not be used for a task if the limitations discussed above are unacceptable or potentially problematic for the inteded use case.

### C.6 Distribution

**Will the dataset be distributed to third parties outside of the entity (e.g., company, institution, organization) on behalf of which the dataset was created?** Yes, the dataset will be made open source and publicly available.

**How will the dataset will be distributed (e.g., tarball on website, API, GitHub)?** The dataset will be distributed via the Hugging Face Hub.

**When will the dataset be distributed?** The dataset will be made available publicly upon publication of this paper.

**Will the dataset be distributed under a copyright or other intellectual property (IP) license, and/or under applicable terms of use (ToU)?** The dataset will be distributed under the CC BY 4.0 license.

**Have any third parties imposed IP-based or other restrictions on the data associated with the instances?** No

**Do any export controls or other regulatory restrictions apply to the dataset or to individual instances?** No

### C.7 Maintenance

**Who will be supporting/hosting/maintaining the dataset?** The datasset will be hosted on the Hugging Face Hub. The authors of this paper will support and maintain the dataset via our public GitHub repository.

**How can the owner/curator/manager of the dataset be contacted (e.g., email address)?** The corresponding author can be contacted via the e-mail address listed on the first page of this paper. Alternatively, an issue can be raised on our GitHub repository.

**Is there an erratum?** No

**Will the dataset be updated (e.g., to correct labeling errors, add new instances, delete instances)?** Although we do not anticipate the need to update this dataset in the future, we will respond to issues which are raised on our public GitHub repository for this project.

**If the dataset relates to people, are there applicable limits on the retention of the data associated with the instances (e.g., were the individuals in question told that their data would be retained for a fixed period of time and then deleted)?** Not applicable

**Will older versions of the dataset continue to be supported/hosted/maintained?** Yes. If the dataset is updated in the future, older versions will remain available.

**If others want to extend/augment/build on/contribute to the dataset, is there a mechanism for them to do so?** Yes, we make our dataset open source and welcome others to build on it. This can be done by making contributions to our GitHub repository and/or citing our dataset as appropriate when used in future work.

