During review, COCO-Counterfactuals and its accompanying code can be accessed via the following link:

https://drive.google.com/drive/folders/1nHKuYCOyU1JH4cNiKa3lNUA4ENvsL51F

This link leads to a Google Drive that includes two folders:

- Folder *COCO-Counterfactuals-Dataset* includes our zipped COCO-Counterfactuals dataset and a README file.
- Folder *COCO-Counterfactuals-SourceCode* includes a zip file and a README file. The zip file includes all of data and implementations that can be used to re-produce our generated COCO-Counterfactuals dataset and experimental results presented in the paper.

While the README file in the former folder describes the structure of our zipped COCO-Counterfactuals dataset, that one in the latter folder details instructions to re-produce our generated COCO-Counterfactuals dataset and experimental results presented in the paper.