# OpenReview forum: "COCO-Counterfactuals: Automatically Constructed Counterfactual Examples for Image-Text Pairs"
_NeurIPS.cc/2023/Track/Datasets_and_Benchmarks — NeurIPS 2023 Datasets and Benchmarks Poster_

### Official Review · Reviewer_BUVv · 2023-07-03
**Human Annotated Counterfactual image-caption dataset with controls based on embedding similarity**

**Rating:** 7
**Confidence:** 4
**Correctness:** The data collection process and evalu…
**Clarity:** The paper is easy to read and the imp…

**Strengths:**

The resulting ~40K dataset when used for data augmentation to train zero-shot models improves OOD performance on some of the test datasets.
The fact that in some of the OOD tasks, using more data hurts, is an interesting observation that can be studied in future work

**Additional Feedback:**

see limitations.

**Documentation:**

The dataset is easy to use and the documentation is sufficient for reproducibility.

**Ethics:**

Limitations and ethical concerns need to be documented in the form of a datasheet.

**Limitations:**

Word substitutions although a scalable manner to generate counterfactuals - may not be realistic. Further, safety checks in the kind of image datasets to not exacerbate gender and racial bias should be studied.

**Opportunities For Improvement:**

The paper can be improved by a thorough error analysis - explaining why there is no improvement in certain datasets, and suggest a way to refine the counterfactual data generation process in a task-specific setting.

**Relation To Prior Work:**

Word substitutions although a scalable manner to generate counterfactuals - may not be realistic - controlled text/image data generation methods can be automatically used in conjunction here, and study of the related work can be improved.

**Summary And Contributions:**

The paper presents a human-annotated counterfactual image-caption dataset with controls based on embedding similarity. The generated pairs are meant to be more diverse based on a range of similarity 0.8-0.91 between text embeddings, while sampled with high-CLIP text-image similarity.

---

> ### Author Response · Authors · 2023-08-21
> **Author Response to Reviewer BUVv**
>
> Thank you for recognizing the utility of our dataset, the clarity of our paper, and the interesting findings of our experiments. We appreciate your insightful comments and have addressed them below.
>
> **The paper can be improved by a thorough error analysis - explaining why there is no improvement in certain datasets.**
>
> Thank you for this suggestion. We added a new analysis of differences in OOD generalization performance on image recognition datasets, which we describe in Appendix A.2:
>
> *“We measured the frequency in which the altered subjects used to produce COCO-Counterfactuals overlapped with class labels. Specifically, we define the COCO-CFs Label Frequency for each image recognition dataset as the total number of COCO-Counterfactuals in which one or more of the dataset’s labels matched one of the two altered subjects used to produce the counterfactual pair….We observe that datasets having a higher COCO-CFs Label Frequency generally achieve larger improvements in OOD generalization performance. The Pearson correlation coefficient between COCO-CFs Label Frequency and the 18 performance change measurements in Table 6 is 0.522 with a p-value of 0.026, indicating statistically significant positive correlation.*
>
> *These results suggest that a major contributor to the variation in OOD generalization performance across datasets is the overlap between the evaluation dataset domain and the set of subjects which are altered in COCO-Counterfactuals. Food101, the only dataset which saw no improvement in performance on our best-performing COCO-CFs training dataset, had only 28 cases of overlap between its label set and the subject alterations in COCO-CFs. In contrast, the greatest performance improvements were achieved on CIFAR100, for which 3446 COCO-CFs had subject alterations matching at least one label from the dataset. These findings point to the potential usefulness of targeting counterfactual changes for task-specific datasets.”*
>
> **The paper can be improved by suggesting a way to refine the counterfactual data generation process in a task-specific setting**
>
> We agree that this is a promising direction for future work and have added the following to Section 6:
>
> *“In this work, we focused on the creation of task-agnostic counterfactual examples. A promising direction for future research is the adaptation of our approach to produce task-specific counterfactuals. For example, in the case of image recognition, the counterfactual changes could be limited to a targeted label distribution to produce counterfactual examples more tailored to the end task. Alternatively, task-specific model failures or spurious correlations could be diagnosed and used as a basis for determining which counterfactual changes to consider when creating the counterfactual captions. We believe that such approaches have the potential to produce counterfactuals which are more targeted for improving specific model deficiencies.”*
>
> **Word substitutions although a scalable manner to generate counterfactuals - may not be realistic - controlled text/image data generation methods can be automatically used in conjunction here, and study of the related work can be improved.**
>
> Thank you for this feedback; we agree with your suggestion and have expanded our discussion of this limitation in Section 7:
>
> *“A motivation for our use of noun substitutions is the desire to produce minimal-edit counterfactuals. This is a common strategy for NLP counterfactuals (Kaushik et al., 2019; Wang and Culotta, 2021; Yang et al., 2021) because the high degree of similarity between the original and counterfactual text preserves spurious correlations that models might rely on for discernment. Furthermore, our use of perplexity filtering mitigates the potential for such word substitutions to produce unrealistic counterfactual captions. However, recent work on NLP counterfactuals have explored alternative generation strategies such as controlled text decoding (Howard et al., 2022), which could be used in lieu of noun substitutions to enable a larger range of counterfactual changes to be considered.”*
>
> **Further, safety checks in the kind of image datasets to not exacerbate gender and racial bias should be studied.**
>
> We agree that this is an important issue to study and have expanded our discussion of it in Section 7 (see also our datasheet in Appendix C):
>
> *“Stable diffusion and other text-to-image diffusion models have been shown to exhibit biases associated with race and gender, including over-representation of masculinity and whiteness (Luccioni et al., 2023); racial and gender disparities in depictions of certain occupations (Bianchi et al., 2023); and preferences for certain genders or skin tones (Cho et al., 2022)...While some recent work has investigated approaches for mitigating biases in diffusion models, further investigation is needed into the de-biasing of datasets on which these models are trained in order to fully eliminate them (Schramowski et al., 2023).”*

---

### Official Review · Reviewer_QvHF · 2023-07-21
**Comments**

**Rating:** 7
**Confidence:** 3
**Correctness:** Yes
**Clarity:** Yes

**Strengths:**

The proposed data augmentation strategy is useful, which is easy to follow and can improve the generalization performance.

The created dataset is also a good testbed for image-text counterfactuals.

**Additional Feedback:**

Please see the above comments

**Documentation:**

Yes

**Ethics:**

Yes

**Limitations:**

please refer to the opportunities for improvement.

**Opportunities For Improvement:**

1.the PDF file is all pixels, and the text and images may be clear after magnifying them.

2.the proposed approach is hard to guarantee the quality and accuracy of the generated images, and human annotators are still required. It would be useful to analyze the found errors and advantages of the diffusion models, to show the confidence boundary of the proposed approach for the generated images.

3.the improvement on the OOD generalization performance seems not significant, more clear explanation about them is required.

**Relation To Prior Work:**

Yes

**Summary And Contributions:**

This paper proposes an automatic data generation method for creating counterfactual examples, to create the paired image-text counterfactuals dataset for existing pre-trained multimodal models and increase the difficulty of zero-shot image-text retrieval and matching tasks. Experimental results show the augmented data can improve the OOD generalization.

---

> ### Author Response · Authors · 2023-08-21
> **Author Response to Reviewer QvHF (Part 1)**
>
> Thank you for recognizing the utility of our COCO-Counterfactuals dataset, the usefulness of our approach for generating image-text counterfactuals, and the clarity of our proposed counterfactual data augmentation strategy. We appreciate your insightful comments and have addressed them below.
>
> **The PDF file is all pixels, and the text and images may be clear after magnifying them.**
>
> We apologize that there was an issue with the PDF that we uploaded for our initial submission. We have fixed this issue in the latest revision, which uses a higher quality PDF with selectable text.
>
> **The proposed approach is hard to guarantee the quality and accuracy of the generated images, and human annotators are still required.**
>
> Thank you for this feedback. While the inability to guarantee the quality and accuracy of automatically generated data is a valid limitation which we discuss in Section 7, we also believe that our approach can be effective even without the use of human annotation. To validate this, we conducted training data augmentation experiments which showed that similar performance is achieved with or without filtering the incorrect COCO-Counterfactuals identified by human annotations (see Section 5.1 and Appendix A.4). While certain use cases which require a high degree of confidence in the accuracy of generated counterfactuals may benefit from the use of human evaluation, we believe that these results demonstrate how our approach can be used for fully automated training data augmentation without human annotation.
>
> We appreciate this comment and have incorporated our above response into Sections 4.1 and 5.1 of the revised manuscript.
>
> **It would be useful to analyze the found errors and advantages of the diffusion models, to show the confidence boundary of the proposed approach for the generated images.**
>
> Thank you for this suggestion. On the one hand, ensuring the quality and accuracy of images generated by diffusion models is a well-known issue. Our approach can utilize any text-to-image diffusion model, which provides flexibility to adapt the latest models which are continuously improving regarding these concerns.
>
> On the other hand, to the best of our knowledge, we are not aware of a theoretical confidence boundary generating images from diffusion models. However, we added an analysis of errors in COCO-Counterfactuals to our latest revision (Appendix A.3). Specifically, we manually categorized 100 sampled image-text pairs which were identified as errors by human annotators.
>
> We found that 66% of the categorized errors were associated with known limitations of models such as Stable Diffusion, including failures to generate a subject/object, fine-grained details, spatial relationships, and the correct number of objects. Additional categories included cases where both prompts were equally valid for an image due to a hyponymy relationship between altered subjects (15%) and human annotation errors (15%). In many cases, these errors do not impact the validity of the depiction of the counterfactual change (i.e., the altered subjects), but rather introduce inaccuracies related to other details in the image. Please see Appendix A.3.1 for a more detailed discussion and examples of the error categories.
>
> In addition to this manual error categorization, we also performed a taxonomic analysis of the altered subjects in COCO-Counterfactuals which were identified as errors by human annotators. We summarize the findings in Appendix A.3.2 as follows:
>
> *"To better understand the relationship between the altered subjects in our counterfactuals and potential failure cases, we conducted a taxonomic analysis of the altered subjects which occurred most frequently among errors identified by human annotators. Table 10 provides the frequency of altered subject pairs which occurred at least 20 times in the error cases identified by human annotators. Interestingly, we observe that 19 of these 20 most frequent altered subject pairs belong to the human taxonomy.*
>
> *We further analyzed this human taxonomy in COCO-Counterfactuals by constructing a list of human-related words, which consists of ‘girl’, ‘boy’, ‘man’, ‘men’, ‘woman’, ‘guy’, ‘kid’, ‘person’, ‘people’, ‘child’, ‘children’, ‘couple’, ‘group’, and ‘lady’. An image-text pair is said to be related to this human taxonomy if the altered subject of its caption belong to this list. We find that there are 4117 image-text pairs in COCO-Counterfactuals that are related to the human taxonomy, among which 1864 were identified as errors by human annotators. The corresponding error rate for altered subjects related to the human taxonomy is 44.3%, which indicates that generating counterfactual pairs involving human altered subjects is more challenging for our approach. This suggests that a promising direction for future work is the exploration of improvements to the generation of images involving human subjects.”*

---

> > ### Author Response · Authors · 2023-08-21
> > **Author Response to Reviewer QvHF (Part 2)**
> >
> > **The improvement on the OOD generalization performance seems not significant, more clear explanation about them is required.**
> >
> > Thank you for this feedback. One promising aspect of our OOD generalization results is that they demonstrate COCO-Counterfactuals are at least as efficient as real image-text sample for training data augmentation, which we describe in the new Section 5.3:
> >
> > *“Recent work investigating the suitability of synthetic training data for image recognition tasks has found that synthetic image data is much less efficient than real data, requiring 5x more synthetic training samples to achieve similar performance as models trained on real data (He et al., 2022). In contrast, our results show that training data augmentation with COCO-Counterfactuals is at least as efficient (Table 3) and sometimes more efficient (Table 4) than data augmentation with an identical amount of real data (|D_train| = 13,928). This finding suggests that our approach could be particularly valuable in low-resource settings where paired image-text data is scarce.”*
> >
> > We have also added the following discussion regarding the magnitude of improvements in OOD generalization performance to Section 5.3:
> >
> > *“While training data augmentation with COCO-Counterfactuals produces statistically significant performance improvements relative to training with only real data, the overall magnitude of these improvements is limited and varies by evaluation setting. COCO-Counterfactuals produce the largest improvements on zero-shot image recognition tasks, where its overall mean improvement over pre trained CLIP is twice as large as that achieved by training on an equivalent amount of real data from MS-COCO. However, OOD generalization performance varies by dataset, which further analysis suggests, is related to domain gaps between altered subjects in COCO-Counterfactuals and the domain of the evaluation dataset (see Appendix A.2 for details)”*
> >
> > Our new analysis in Appendix A.2 describes differences in OOD generalization performance on image recognition datasets as follows:
> >
> > *“We measured the frequency in which the altered subjects used to produce COCO-Counterfactuals overlapped with class labels. Specifically, we define the COCO-CFs Label Frequency for each image recognition dataset as the total number of COCO-Counterfactuals in which one or more of the dataset’s labels matched one of the two altered subjects used to produce the counterfactual pair….We observe that datasets having a higher COCO-CFs Label Frequency generally achieve larger improvements in OOD generalization performance. The Pearson correlation coefficient between COCO-CFs Label Frequency and the 18 performance change measurements in Table 6 is 0.522 with a p-value of 0.026, indicating statistically significant positive correlation.*
> >
> > *These results suggest that a major contributor to the variation in OOD generalization performance across datasets is the overlap between the evaluation dataset domain and the set of subjects which are altered in COCO-Counterfactuals. Food101, the only dataset which saw no improvement in performance on our best-performing COCO-CFs training dataset, had only 28 cases of overlap between its label set and the subject alterations in COCO-CFs. In contrast, the greatest performance improvements were achieved on CIFAR100, for which 3446 COCO-CFs had subject alterations matching at least one label from the dataset. These findings point to the potential usefulness of targeting counterfactual changes for task-specific datasets.”*
> >
> > Finally, our new evaluations in Appendix A.1 suggest that COCO-Counterfactuals may offer greater improvements to the robustness of models to minimal/counterfactual changes in images:
> >
> > *“By design, COCO-Counterfactuals may offer greater improvements to the robustness of models to minimal or counterfactual changes in images. Such examples are unlikely to be present in the datasets used previously to evaluate OOD generalization. Therefore, we also evaluate the performance of models on a withheld test set of COCO-Counterfactuals to determine their image-text retrieval capabilities on in-domain counterfactual examples…We observe that training on COCO-Counterfactuals results in a mean improvement of 11.83, 21.55, and 11.47 relative to the pre-trained CLIP, BridgeTower, and Flava models, respectively. This represents an average relative improvement of 24.3% for each model over the performance of its pre-trained version. In addition, the CLIP, BridgeTower, and Flava models that were trained on COCO-Counterfactuals achieve a mean absolute improvement of 6.06, 10.08, and 5.28, respectively, relative to those that were trained on MS-COCO. The greater magnitude of these performance gains relative to our OOD image-text retrieval evaluations (Table 3) suggests that training on COCO-Counterfactuals improves model robustness to counterfactual changes, which are not present in our (non-counterfactual) OOD evaluation datasets.”*

---

> > > ### Comment · Reviewer_QvHF · 2023-08-30
> > > **Response and Raise Score**
> > >
> > > Thank you to the authors for answering all the questions. Upon reviewing their responses, I raised my original scores.

---

### Official Review · Reviewer_jADd · 2023-07-21
**COCO-Counterfactuals: Automatically Constructed Counterfactual Examples for Image-Text Pairs**

**Rating:** 8
**Confidence:** 3
**Correctness:** This sounds correct.
**Clarity:** Yes

**Strengths:**

This automated counter factual generation improves both evaluation of current models, as well as actual training dataset quantity.

On the evaluation side, counter factual generations provide a harder test. Table 2 shows that current models are finding this task challenging.

On the training data side, it is able to provide more samples, and thus allow for larger training dataset. The authors provide extensive experiments in order to measure general improvements of those additional samples in the training dataset with even variance estimation thus showing some statistical relevance to the given results

**Additional Feedback:**

None

**Documentation:**

Yes, the process for dataset creation is detailed enough. Everything seems to be built on top of MSCOCO

**Limitations:**

Using noun replacement as counter factual generation mechanisms sounds very limiting, especially sine there are strong generative systems that can provide a higher variance of text.

**Opportunities For Improvement:**

It's unclear that the additional samples improve the model given a specific dataset size. Looking at Table 3, it's unclear that given a training dataset size, the mixture improves over pure MS COCO dataset. The table 4 does provide statistically meaningful improvement, yet some results are curious. Typically that the bigger the training dataset, the worse the model becomes. It would have been interesting to verify the 50-50 rule in the mixture. Additionally, if the goal is to provide more dataset samples, it would be valuable to compare with training runs with only MS COCO with multiple epochs.

Also interestingly, it's my understanding that the original dataset does not have to paired. One could generate those counter factual examples given a single text, or a single image.

**Relation To Prior Work:**

I am not an expert on the field, but it does mention that:
 - it added images associated to the counter factual text, despite using a similar "foil" system

**Summary And Contributions:**

Contributions:
 - generate convert factual examples for image-text pair tasks in an automated way
 - evaluation of the difficulty of those samples in zero-shot setting. Typically those counter-factual examples make for more challenging benchmarks
 - generated examples can be used ad additional training samples.

---

> ### Author Response · Authors · 2023-08-21
> **Author Response to Reviewer jADd**
>
> Thank you for recognizing the usefulness of our method, the utility of COCO-Counterfactuals for training data augmentation & model evaluation, and the extensiveness of our experiments. We appreciate your insightful comments and have addressed them below.
>
> **It's unclear that the additional samples improve the model given a specific dataset size. Looking at Table 3, it's unclear that given a training dataset size, the mixture improves over pure MS COCO dataset.**
>
> Thank you for this this observation. Our aim is not necessarily to show that COCO-Counterfactuals are a replacement for real data, but that their efficiency is comparable to real data and that they can improve the OOD generalization performance of models when used to augment real training datasets. We have added the following new discussion to Section 5.3 to elaborate on the efficiently of COCO-Counterfactuals relative to real data:
>
> *“Recent work investigating the suitability of synthetic training data for image recognition tasks has found that synthetic image data is much less efficient than real data, requiring 5x more synthetic training samples to achieve similar performance as models trained on real data (He et al., 2022). In contrast, our results show that training data augmentation with COCO-Counterfactuals is at least as efficient (Table 3) and sometimes more efficient (Table 4) than data augmentation with an identical amount of real data (|D_train| = 13,928). This finding suggests that our approach could be particularly valuable in low-resource settings where paired image-text data is scarce.”*
>
> **The table 4 does provide statistically meaningful improvement, yet some results are curious. Typically that the bigger the training dataset, the worse the model becomes.**
>
> We appreciate this feedback and have incorporated the following discussion on the impact of counterfactual augmentation size to the new Section 5.3 in our latest revision:
>
> *“Consistent with prior work on training data augmentation with NLP counterfactuals (Howard et al., 2022; Joshi and He, 2022), Tables 3 and 4 show that improvements in OOD performance with increasing amounts of counterfactual examples reaches a saturation point, beyond which additional data augmentation does not lead to further improvements. For image-text retrieval on Flickr30k (Table 3), this saturation point is reached with a 40 / 60% mixture of MS-COCO / COCO-Counterfactuals in the training dataset. In contrast, Table 4 shows that the saturation point for the OOD image recognition datasets is reached with a 50 / 50% split based on the mean of the six datasets. These results suggest that the optimal mixture of real examples and synthetically generated counterfactual examples may differ depending on the evaluation task and dataset."*
>
> **It would have been interesting to verify the 50-50 rule in the mixture. If the goal is to provide more dataset samples, it would be valuable to compare with training runs with only MS COCO with multiple epochs.**
>
> Regarding training dataset mixture, the third row of Tables 3 & 4 provides the result of training on a 50% / 50% mixture of MS-COCO / COCO-Counterfactuals. To make this clearer, we added a new third column to the table that indicates the percentage of the training dataset which is comprised of COCO-Counterfactuals. We also provide the result of training only on MS-COCO in row 2 of Tables 3 & 4.
>
> **It's my understanding that the original dataset does not have to paired. One could generate those counter factual examples given a single text, or a single image.**
>
> Yes, this is correct – an advantage of our approach is that it can produce a paired image-text dataset of counterfactual examples without access to an original image-text dataset. We appreciate you pointing this out and have added this comment to Section 3.3.
>
> **Using noun replacement as counter factual generation mechanisms sounds very limiting, especially sine there are strong generative systems that can provide a higher variance of text.**
>
> Thank you for this feedback. We have expanded our discussion of this limitation and the potential of applying alternative generative methods in Section 7:
>
> *“A motivation for our use of noun substitutions is the desire to produce minimal-edit counterfactuals. This is a common strategy for NLP counterfactuals (Kaushik et al., 2019; Wang and Culotta, 2021; Yang et al., 2021) because the high degree of similarity between the original and counterfactual text preserves spurious correlations that models might rely on for discernment. Furthermore, our use of perplexity filtering mitigates the potential for such word substitutions to produce unrealistic counterfactual captions. However, recent work on NLP counterfactuals have explored alternative generation strategies such as controlled text decoding (Howard et al., 2022), which could be used in lieu of noun substitutions to enable a larger range of counterfactual changes to be considered.”*

---

### Official Review · Reviewer_ndME · 2023-07-31

**Rating:** 7
**Confidence:** 3
**Correctness:** Yes
**Clarity:** Yes

**Strengths:**

- To the best of my knowledge, the contribution of an image-text paired counterfactual dataset is novel and interesting to the community.
- The annotation process is clearly documented.
- The paper’s execution on human-machine collaborative data annotation could inspire future efforts.


**Additional Feedback:**

N/A

**Documentation:**

Yes

**Limitations:**

The paper includes a decent discussion of the limitations. I would appreciate it if the authors can further discuss the undesired artifacts that might be brought about by heavily relying on pretrained models in the annotation process.

**Opportunities For Improvement:**

- Despite the opening motivations, the paper does not include any in-depth discussion about the spurious correlations that COCO-counterfactuals helps reveal.
- Little evidence is provided on how the proposed dataset can help improve/evaluate the models’ robustness.
- Some of the claims are questionable. E.g., in Section 5.1 and Table 3, I don’t find the improvements by training on COCO-counterfactuals “significant” based on the stddev numbers. I’d suggest toning down.


**Relation To Prior Work:**

Yes

**Summary And Contributions:**

This paper introduces a new counterfactual dataset for image-text paired data. It is inspired by the counterfactual data in NLP, and introduces perturbations to image-text pairs to quantify the models’ reliance on spurious correlations as evaluation data, and improve the models robustness through data augmentation. The annotation process builds on the collaboration between human annotators and recent pretrained image and text generation models. Extensive experiments are conducted on the proposed dataset, leading to interesting findings.

---

> ### Author Response · Authors · 2023-08-21
> **Author Response to Reviewer ndME (Part 1)**
>
> Thank you for recognizing the novelty of our counterfactual image-text dataset, the clarity of documentation in our paper, its ability to inspire future efforts on human-machine collaborative data annotation, and the extensiveness of our experiments. We appreciate your insightful comments and have addressed them below.
>
> **Despite the opening motivations, the paper does not include any in-depth discussion about the spurious correlations that COCO-counterfactuals helps reveal.**
>
> Our work was inspired by the idea that training models with counterfactual examples provides a strong inductive bias against learning spurious correlations in datasets, leading to greater robustness and improved generalization on OOD data (Eisentein, 2022; Vig et al. 2020). While our intent was not to introduce COCO-Counterfactuals as a medium to reveal or diagnose specific spurious correlations that a pre-trained model has learned, we believe that this idea is an interesting direction for future research. Indeed, such revealed spurious correlations could inform how to best generate counterfactual examples to target specific model failures. We appreciate this feedback and have added the following new discussion of this opportunity to Section 6:
>
> *“Alternatively, task-specific model failures or spurious correlations could be diagnosed and used as a basis for determining which counterfactual changes to consider when creating the counterfactual captions. We believe that such approaches have the potential to produce counterfactuals which are more targeted for improving specific model deficiencies.”*
>
> **Little evidence is provided on how the proposed dataset can help improve/evaluate the models’ robustness.**
>
> Thank you for this feedback. While we considered a comprehensive study of model robustness to be beyond the scope of this work, our training data augmentation experiments were designed to partially investigate robustness through the use of out-of-domain (OOD) test datasets. For example, the image-text retrieval results reported in Table 3 and zero-shot classification accuracy results reported in Table 4 all utilize evaluation datasets which are OOD for the trained models. We believe that the statistically significant improvements on these OOD datasets achieved by models trained on COCO-counterfactuals provides evidence of our dataset’s ability to improve model robustness. To clarify the intention of our use of OOD datasets, we have added the following sentence to Section 5:
>
> *“In order to investigate the robustness of models trained on COCO-CFs, we evaluate them on OOD datasets for image-text retrieval and image recognition.”*
>
> Additionally, we have added the following new paragraph to Section 4.2.3 which discusses how our zero-shot testing of existing pre-trained models on COCO-Counterfactuals demonstrates their utility for evaluating model robustness to counterfactual changes in image-text data:
>
> *“When used as a test set, COCO-Counterfactuals by design evaluate the robustness of models to minimal changes in paired image-text data. Table 2 and Figure 3 show that existing models perform significantly worse when evaluated on COCO-Counterfactuals. Additionally, we find that training these same models on COCO-Counterfactuals produces an average relative improvement of 24.3% in image-text retrieval performance on withheld counterfactual examples (see Table 5 of Appendix A.1). These results point to the usefulness our dataset for evaluating and improving the robustness of multimodal models to counterfactual changes.”*
>
> Finally, our new evaluations in Appendix A.1 suggest that COCO-Counterfactuals may offer greater improvements to the robustness of models to minimal/counterfactual changes in images:
>
> *“By design, COCO-Counterfactuals may offer greater improvements to the robustness of models to minimal or counterfactual changes in images. Such examples are unlikely to be present in the datasets used previously to evaluate OOD generalization. Therefore, we also evaluate the performance of models on a withheld test set of COCO-Counterfactuals to determine their image-text retrieval capabilities on in-domain counterfactual examples…We observe that training on COCO-Counterfactuals results in a mean improvement of 11.83, 21.55, and 11.47 relative to the pre-trained CLIP, BridgeTower, and Flava models, respectively. This represents an average relative improvement of 24.3% for each model over the performance of its pre-trained version. In addition, the CLIP, BridgeTower, and Flava models that were trained on COCO-Counterfactuals achieve a mean absolute improvement of 6.06, 10.08, and 5.28, respectively, relative to those that were trained on MS-COCO. The greater magnitude of these performance gains relative to our OOD image-text retrieval evaluations (Table 3) suggests that training on COCO-Counterfactuals improves model robustness to counterfactual changes, which are not present in our (non-counterfactual) OOD evaluation datasets.”*

---

> > ### Author Response · Authors · 2023-08-21
> > **Author Response to Reviewer ndME (Part 2)**
> >
> > **Some of the claims are questionable. E.g., in Section 5.1 and Table 3, I don’t find the improvements by training on COCO-counterfactuals “significant” based on the stddev numbers. I’d suggest toning down.**
> >
> > Thank you for this suggestion. Our discussion of the significant improvement in Table 3 pertained to that of models trained on COCO-Counterfactuals relative to pre-trained CLIP (approx. 5-point mean improvement with a standard deviation of 0.2). However, we agree with your comment and have replaced our previous statement in Section 5.1 that our models “significantly outperform” with the following more precise statement which expresses the magnitude of the performance difference:
> >
> > *“We observe that all CLIP models trained with COCO-Counterfactuals outperform pre-trained CLIP by an average of 5 points, based on the mean performance across text and image retrieval settings.”*
> >
> > Additionally, we added the following discussion regarding the magnitude of improvements in OOD generalization performance to Section 5.3:
> >
> > *“While training data augmentation with COCO-Counterfactuals produces statistically significant performance improvements relative to training with only real data, the overall magnitude of these improvements is limited and varies by evaluation setting. COCO-Counterfactuals produce the largest improvements on zero-shot image recognition tasks, where its overall mean improvement over pre-trained CLIP is twice as large as that achieved by training on an equivalent amount of real data from MS-COCO. However, OOD generalization performance varies by dataset, which further analysis suggests, is related to domain gaps between altered subjects in COCO-Counterfactuals and the domain of the evaluation dataset (see Appendix A.2 for details)”*
> >
> > **The paper includes a decent discussion of the limitations. I would appreciate it if the authors can further discuss the undesired artifacts that might be brought about by heavily relying on pretrained models in the annotation process.**
> >
> > We appreciate this suggestion and have added the following paragraph to the discussion of limitations in Section 7:
> >
> > *“Despite the impressive recent improvements in text-to-image generation capabilities, models such as Stable Diffusion have well-known limitations that should be considered when utilizing datasets which are derived from them. These limitations include unrealistic depictions of hands, palms, and other fine-grained objects (Samuel et al., 2023); failures to generate one or more of the subjects in a prompt and correctly bind attributes such as color (Chefer et al., 2023); difficulties with object counting and spatial relationship understanding (Cho et al., 2022); and challenges associated with the composition of concepts (Liu et al., 2022). While our experiments suggest that COCO-Counterfactuals is relatively robust to generation failures when used for training data augmentation, future applications of our methodology should consider the risks associated with these limitations relative to the intended use of the generated dataset.”*

---

### Author Response · Authors · 2023-08-21
**General Response to All Reviewers**

Dear reviewers,

Thank you for taking the time to evaluate our paper and provide valuable feedback on it, which has helped us make substantial improvements in our latest draft. We appreciate your recognition of the novelty of our counterfactual image-text dataset (Reviewer ndME) as well as its utility for training data augmentation and model evaluation (Reviewers QvHF, jADd, & BUVv); the usefulness of our automated method for generating image-text counterfactuals (Reviewers jADd & QvHF); the clarity of the methods described in our paper (Reviewers ndME, QvHF, & BUVv); and the extensiveness of our experiments (Reviewers ndME & jADd) as well as the interesting findings they produced (Reviewer BUVv).

Our revised paper contains significant additions in response to your comments. We summarize major additions below and highlight all new content in blue text in our latest draft:

1.	Evaluation of robustness to counterfactual changes after training CLIP, BridgeTower, and Flava on COCO-Counterfactuals (Appendix A.1)
2.	Analysis of differences in OOD generalization performance on image recognition datasets (Appendix A.2)
3.	Analysis of errors in COCO-Counterfactuals identified by human annotators (Appendix A.3)
4.	Discussion of the usefulness of COCO-Counterfactuals for evaluating model robustness (Section 4.2.3)
5.	Discussion of OOD generalization performance after training on COCO-Counterfactuals (Section 5.3)
6.	Discussion of opportunities for future work on creating task-specific counterfactuals (Section 6)
7.	Expanded discussion of limitations and ethical concerns (Section 7)

For ease of discussion, we have also quoted relevant segments of our new additions directly in our responses to your individual reviews. We greatly appreciate your valuable feedback on our paper and hope we have addressed your concerns; please let us know if you have any additional questions regarding our latest revision.

Sincerely,

Submission 573 Authors

---

### Decision · Program_Chairs · 2023-09-22

**Decision:**

Accept (Poster)

**Comment:**

This paper introduces an automated generation of counterfactual examples from image-text pairs and creates COCO-CFs from MS-CoCo dataset. Experimental results demonstrate that this dataset can increase the difficulty of zero-shot image-text retrieval task.

**strengths**

* Easy and effective data augmentation method that can be useful for both evaluation and training.
* The contribution of an image-text paired counterfactual dataset is novel and interesting to the community.

**weaknesses/suggestions**

* No clear gains in ood generalization
* No safety checks  to detect gender and racial bias

I think the paper makes a nice contribution that the community will find valuable.  However, I encourage the authors to think carefully about how to reflect the comments or resolve the questions from reviewers in the camera ready version.